# From Similarity to Superiority: Channel Clustering for Time Series Forecasting

**Jialin Chen[1], Jan Eric Lenssen[2,3], Aosong Feng[1], Weihua Hu[2],**
**Matthias Fey[2], Leandros Tassiulas[1], Jure Leskovec[2,4], Rex Ying[1]**
[1]Yale University, [2]Kumo.AI,
[3]Max Planck Institute for Informatics, [4]Stanford University

## Abstract

Time series forecasting has attracted significant attention in recent decades. Previous studies have demonstrated that the Channel-Independent (CI) strategy improves forecasting performance by treating different channels individually, while it leads to poor generalization on unseen instances and ignores potentially necessary interactions between channels. Conversely, the Channel-Dependent (CD) strategy mixes all channels with even irrelevant and indiscriminate information, which, however, results in oversmoothing issues and limits forecasting accuracy. There is a lack of channel strategy that effectively balances individual channel treatment for improved forecasting performance without overlooking essential interactions between channels. Motivated by our observation of a correlation between the time series model's performance boost against channel mixing and the intrinsic similarity on a pair of channels, we developed a novel and adaptable **C**hannel **C**lustering **M**odule (CCM). CCM dynamically groups channels characterized by intrinsic similarities and leverages cluster information instead of individual channel identities, combining the best of CD and CI worlds. Extensive experiments on real-world datasets demonstrate that CCM can (1) boost the performance of CI and CD models by an average margin of $2.4\%$ and $7.2\%$ on long-term and short-term forecasting, respectively; (2) enable zero-shot forecasting with mainstream time series forecasting models; (3) uncover intrinsic time series patterns among channels and improve interpretability of complex time series models [1].

## 1 Introduction

Time series forecasting has attracted a surge of interest across diverse fields, ranging from economics, energy [1, 2], weather [3, 4], to transportation planning [5, 6]. The complexity of the task is heightened by factors including seasonality, trend, noise in the data, and potential cross-channel information.

Despite the numerous deep learning time series models proposed recently [7, 8, 9, 10, 11, 12, 13, 14], an unresolved challenge persists in the effective management of channel interaction within the forecasting framework [15, 16]. Previous works have explored two primary channel strategies: Channel-Independent (CI) and Channel-Dependent (CD) strategies. The Channel-Independent (CI) strategy has shown promise in better forecasting performance by having individual models for each channel. However, a critical drawback is its limited generalizability and robustness on unseen channels [17]. Besides, it tends to overlook potential interactions between various channels. Conversely, the Channel-Dependent (CD) strategy models all channels as a whole and captures intricate channel relations, while they tend to show oversmoothing and have trouble fitting to individual channels, especially when the similarity between channels is very low. Moreover, existing

---

[1]The code is available at `https://github.com/Graph-and-Geometric-Learning/TimeSeriesCCM`

38th Conference on Neural Information Processing Systems (NeurIPS 2024).

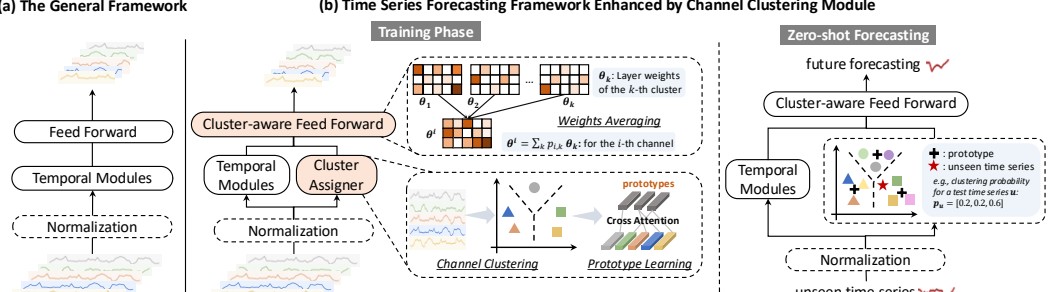

Figure 1: The pipeline of applying Channel Clustering Module (CCM) to general time series models. (a) is the general framework of most time series models. (b) illustrates two modified modules when applying CCM: Cluster Assigner and Cluster-aware Feed Forward. Cluster Assigner learns channel clustering based on intrinsic similarities and creates prototype embeddings for each cluster via a cross-attention mechanism. The clustering probabilities $\{p_{i,k}\}$ are subsequently used in Cluster-aware Feed Forward to average $\{\theta_k\}_{k=1}^K$, which are layer weights assigned to $K$ clusters, obtaining weights $\theta^i$ for the $i$-th channel. The learned prototypes retain pre-trained knowledge, enabling zero-shot forecasting on unseen samples in both univariate and multivariate scenarios.

models typically treat univariate data in a CI manner, neglecting the interconnections between time series samples, even though these dependencies are commonly observed and beneficial in real-world scenarios, such as stock market or weather forecasting [18, 19, 20].

**Proposed work**. To address the aforementioned challenges, we propose a Channel Clustering Module (CCM) that balances individual channel treatment and captures necessary cross-channel dependencies simultaneously. CCM is motivated by the key observations that CI and CD models typically rely on channel identity information. The level of reliance is anti-correlated with the similarity between channels (see Sec. 4.1 for an analysis). This intriguing phenomenon alludes to the model's analogous behavior on similar channels. The proposed CCM thereby involves the strategic clustering of channels into cohesive clusters, where intra-cluster channels exhibit a higher degree of similarity. To capture the underlying time series patterns within these clusters, we employ cluster-aware Feed Forward to assign independent weights to each cluster and replace individual channel treatment with individual cluster treatment. Moreover, CCM learns expressive prototype embeddings in training, which enables zero-shot forecasting on unseen samples by grouping them into appropriate clusters.

CCM is a plug-and-play solution that is adaptable to most mainstream time series models. We evaluate the effectiveness of CCM on four different time series backbones (*aka.* base models): TSMixer [7], DLinear [8], PatchTST [21], and TimesNet [13]. It can also be applied to other state-of-the-art models for enhanced performance. Extensive experiments verify the superiority of CCM in long-term and short-term forecasting benchmarks, achieving an average margin of 2.4% and 7.2%, respectively. Additionally, we collect stock data from a diverse range of companies to construct a new stock univariate dataset. Leveraging information from intra-cluster samples, CCM consistently shows a stronger ability to accurately forecast stock prices in the dynamic and intricate stock market. Moreover, CCM enhances zero-shot forecasting capacities of time series backbones in cross-domain scenarios, which further highlights the robustness and versatility of CCM.

The **contributions** of this paper are: (1) We propose a novel and unified channel strategy, *i.e.*, CCM, which is adaptable to most mainstream time series models. CCM explores the optimal trade-off between channel individual treatment and cross-channel modeling, (2) CCM demonstrates superiority in improving performance on long-term and short-term forecasting, and (3) through learning prototypes from clusters, CCM enables zero-shot forecasting on unseen samples in both univariate and multivariate scenarios.

## 2 Related Work

### 2.1 Time Series Forecasting Models

Traditional machine learning methods such as Prophet [22, 23], ARIMA [24] capture the trend component and seasonality in time series [25]. As data availability continues to grow, deep learning

methods revolutionized this field, introducing more complex and efficient models [26, 27]. Convolutional Neural Networks (CNNs) [13, 14, 28, 29, 30], have been widely adopted to capture local temporal dependencies. Recurrent Neural Networks (RNNs) [31, 32, 33, 34, 28] excel in capturing sequential information, yet they often struggle with longer sequences. Transformer-based models [11, 35, 12, 21, 36, 37, 9, 38, 10, 39, 40], typically equipped with self-attention mechanisms [41], demonstrate their proficiency in handling long-range dependencies, although they require substantial computational resources. Recently, linear models [42, 43, 44], *e.g.,* DLinear [8], TSMixer [7], have gained popularity for their simplicity and effectiveness in long-term time series forecasting, but they may underperform with non-linear and complex patterns. Besides, traditional tricks, including trend-seasonal decomposition [8, 45, 46] and multi-periodicity analysis [47, 48, 13, 49, 50, 51, 52] continue to play a crucial role in aiding in the preprocessing stage for advanced models.

## 2.2 Channel Strategies in Time Series Forecasting

Most deep learning models [12, 39, 10] adopt the Channel-Dependent (CD) strategy, aiming to harness the full spectrum of information across channels. Conversely, the Channel-Independent (CI) approaches [21, 8] build forecasting models for each channel independently. Prior works on CI and CD strategy [17, 15, 53, 54, 16] present that CI leads to higher capacity and lower robustness, whereas CD is the opposite. Predicting residuals with regularization (PRReg) [17] is thereby proposed to incorporate a regularization term in the objective to encourage smoothness in future forecasting. However, the essential challenge from the model design perspective has not been solved and it remains challenging to develop a balanced channel strategy. Prior research has explored effective clustering of channels to improve the predictive capabilities in diverse applications, including image classification [55], natural language processing (NLP) [56, 57], anomaly detection [58, 59, 60]. For instance, in traffic prediction [61, 62], clustering techniques have been proposed to group relevant traffic regions to capture intricate spatial patterns. Despite the considerable progress in these areas, the potential and effect of channel clustering in time series forecasting remain under-explored.

## 3 Preliminaries

**Time Series Forecasting**. Formally, let $X = [\boldsymbol{x}_1, \ldots \boldsymbol{x}_T] \in \mathbb{R}^{T \times C}$ be a time series, where $T$ is the length of historical data. $\boldsymbol{x}_t \in \mathbb{R}^C$ represents the observation at time $t$. $C$ denotes the number of variates (*i.e.,* channels). The objective is to construct a predictive model $f$ that estimates the future values of the series, $Y = [\hat{\boldsymbol{x}}_{T+1}, \ldots, \hat{\boldsymbol{x}}_{T+H}] \in \mathbb{R}^{H \times C}$, where $H$ is the forecasting horizon. We use $X_{[:,i]} \in \mathbb{R}^T$ ($X_i$ for simplicity) to denote the $i$-th channel in the time series.

**Channel Dependent (CD) and Channel Independent (CI)**. The CI strategy models each channel $X_i$ separately and ignores any potential cross-channel interactions. This approach is typically denoted as $f^{(i)} : \mathbb{R}^T \to \mathbb{R}^H$ for $i = 1, \cdots, C$, where $f^{(i)}$ is specifically dedicated to the $i$-th channel. Refer to Appendix A.2 for more details. In contrast, the CD strategy models all the channels as a whole with a function $f : \mathbb{R}^{T \times C} \to \mathbb{R}^{H \times C}$. This strategy is essential in scenarios where channels are not just parallel data streams but are interrelated, such as in financial markets or traffic flows.

## 4 Proposed Method

In this work, we propose a Channel Clustering Module (CCM), a model-agnostic method that is adaptable to most mainstream time series models. The pipeline of applying CCM is visualized in Figure 1. General time series models, shown in Figure 1(a), typically consist of three core components [15, 63]: an optional normalization layer (*e.g.,* RevIN [64], SAN [65]), temporal modules including linear layers, transformer-based, or convolutional backbones, and a feed-forward layer that forecasts the future values. Motivated by the empirical observation discussed in Sec. 4.1, CCM presents with a cluster assigner preceding the temporal modules, followed by a cluster-aware Feed Forward (Sec. 4.2). The cluster assigner implements channel clustering based on intrinsic similarities and employs a cross-attention mechanism to generate prototypes for each cluster, which stores the knowledge from the training set and endows the model with zero-shot forecasting capacities.

## 4.1 Motivation for Channel Similarity

To motivate our similarity-based clustering method, we conduct the following toy experiment. We select four recent and popular time series models with different backbones. TSMixer [7] and DLinear are linear models. PatchTST [21] is a transformer-based model with a patching mechanism and TimesNet [13] is a convolutional network that captures multi-periodicity in data. Among these, TSMixer and TimesNet utilize a Channel-Dependent strategy while DLinear and PatchTST adopt the Channel-Independent design. We train a time series model across all channels and evaluate the channel-wise Mean Squared Error (MSE) loss on the test set. Then, we repeat training while randomly shuffling channels in each batch. Note that for both CD and CI models, this means channel identity information will be removed. We report the average performance gain in terms of MSE loss across all channels based on the random shuffling experiments (denoted as $\Delta\mathcal{L}(\%)$) in Table 1.

We attribute the models' performance decrease in the random shuffling experiments to the loss of *channel identity information*. We see that all models rely on channel identity information to achieve better performance. Next, we define channel similarity based on radial basis function kernels [66] as

$$\mathrm{SIM}(X_i, X_j) = \exp(\frac{-\|X_i - X_j\|^2}{2\sigma^2}), \quad (1)$$

where $\sigma$ is a scaling factor. Note that the similarity is computed on the standardized time series to avoid scaling differences. More details are discussed in Appendix A.1. The performance

Table 1: Averaged performance gain from channel identity information ($\Delta\mathcal{L}(\%)$) and Pearson Correlation Coefficients (*PCC*) between $\{\Delta\mathcal{L}_{ij}\}_{i,j}$ and $\{\mathrm{SIM}(X_i, X_j)\}_{i,j}$. The values are averaged across all test samples.

| Base Model Channel Strategy | | TSMixer CD | DLinear CI | PatchTST CI | TimesNet CD |
|---|---|---|---|---|---|
| **ETTh1** | $\Delta\mathcal{L}(\%)$ | 2.67 | 1.10 | 11.30 | 18.90 |
| | PCC | - 0.67 | - 0.66 | - 0.61 | - 0.66 |
| **ETTm1** | $\Delta\mathcal{L}(\%)$ | 4.41 | 5.55 | 6.83 | 14.98 |
| | PCC | - 0.68 | - 0.67 | - 0.68 | - 0.67 |
| **Exchange** | $\Delta\mathcal{L}(\%)$ | 16.43 | 19.34 | 27.98 | 24.57 |
| | PCC | - 0.62 | - 0.62 | - 0.47 | - 0.49 |

difference in MSE from the random shuffling experiment for channel $i$ is denoted as $\Delta\mathcal{L}_i$. We define $\Delta\mathcal{L}_{ij} := |\Delta\mathcal{L}_i - \Delta\mathcal{L}_j|$ and calculate the Pearson Correlation Coefficients (PCC) between $\{\Delta\mathcal{L}_{ij}\}_{i,j}$ and $\{\mathrm{SIM}(X_i, X_j)\}_{i,j}$, as shown in Table 1. The toy example verifies the following two assumptions: **(1)** Existing forecasting methods heavily rely on channel identity information. **(2)** This reliance clearly anti-correlates with channel similarity: for channels with high similarity, channel identity information is less important. Together, these two assumptions motivate us to design an approach that provides cluster identity instead of channel identity, combining the best of both worlds: high capacity and generalizability.

## 4.2 CCM: Channel Clustering Module

**Channel Clustering**. Motivated by the above observations, we first initialize a set of $K$ cluster embeddings $\{c_1, \cdots, c_K\}$, where $c_k \in \mathbb{R}^d$, $d$ is the hidden dimension and $K$ is a hyperparameter. Given a multivariate time series $X \in \mathbb{R}^{T \times C}$, each channel in the input $X_i$ is transformed into a $d$-dimensional channel embedding $h_i$ through an MLP. The probability that a given channel $X_i$ is associated with the $k$-th cluster is the normalized inner-product of the cluster embedding $c_k$ and the channel embedding $h_i$, which is computed as

$$p_{i,k} = \mathrm{Normalize}(\frac{c_k^\top h_i}{\|c_k\|\|h_i\|}) \in [0, 1]. \quad (2)$$

The normalization operator ensures that $\sum_k p_{i,k} = 1$ and validates the clustering probability distribution across $k$ clusters. We utilize reparameterization trick [67] to obtain the clustering membership matrix $\mathbf{M} \in \mathbb{R}^{C \times K}$ where $\mathbf{M}_{ik} \approx \mathrm{Bernoulli}(p_{i,k})$. Higher probability $p_{i,k}$ results in $\mathbf{M}_{ik}$ close to 1, leading to the deterministic existence of certain channels in the corresponding cluster.

**Prototype Learning**. The cluster assigner also creates a $d$-dimensional prototype embedding for each cluster in the training phase. Let $\mathbf{C} = [c_1, \cdots, c_K] \in \mathbb{R}^{K \times d}$ denote the cluster embedding, and $\mathbf{H} = [h_1, \cdots, h_C] \in \mathbb{R}^{C \times d}$ denote the hidden embedding of the channels. To emphasize the intra-cluster channels and remove interference from out-of-cluster channel information, we design a modified cross-attention as follows,

$$\widehat{\mathbf{C}} = \mathrm{Normalize}\left(\exp(\frac{(W_Q\mathbf{C})(W_K\mathbf{H})^\top}{\sqrt{d}}) \odot \mathbf{M}^\top\right) W_V\mathbf{H}, \quad (3)$$

where the clustering membership matrix $\mathbf{M}$ is an approximately binary matrix to enable sparse attention on intra-cluster channels specifically. $W_Q$, $W_K$ and $W_V$ are learnable parameters. The prototype embedding $\widehat{\mathbf{C}} \in \mathbb{R}^{K \times d}$ serves as the updated cluster embedding for subsequent clustering probability computing in Eq. 2.

**Cluster Loss**. We further introduce a specifically designed loss function for the clustering quality, termed ClusterLoss, which incorporates both the alignment of channels with respective clusters and the distinctness between different clusters in a self-supervised context. Let $\mathbf{S} \in \mathbb{R}^{C \times C}$ denote the channel similarity matrix $\mathbf{S}_{ij} = \text{SIM}(X_i, X_j)$ defined in Eq. 1. The ClusterLoss is formulated as:

$$\mathcal{L}_C = -\text{Tr}\left(\mathbf{M}^\top \mathbf{S} \mathbf{M}\right) + \text{Tr}\left(\left(\mathbf{I} - \mathbf{M}\mathbf{M}^\top\right)\mathbf{S}\right), \tag{4}$$

where Tr indicates a trace operator. $\text{Tr}\left(\mathbf{M}^\top \mathbf{S}\mathbf{M}\right)$ maximizes the channel similarities within clusters, which is a fundamental requirement for effective clustering. $\text{Tr}\left(\left(\mathbf{I} - \mathbf{M}\mathbf{M}^\top\right)\mathbf{S}\right)$ instead encourages separation between clusters, which further prevents overlap and ambiguity in clustering assignments. $\mathcal{L}_C$ captures meaningful time series prototypes without relying on external labels or annotations. The overall loss function thereby becomes $\mathcal{L} = \mathcal{L}_F + \beta\mathcal{L}_C$, where $\mathcal{L}_F$ is the general forecasting loss such as MSE loss; and $\beta$ is a regularization parameter for a balance between forecasting accuracy and cluster quality.

**Cluster-aware Feed Forward**. Instead of using individual Feed Forward per channel in a CI manner or sharing one Feed Forward across all channels in a CD manner, we assign a separate Feed Forward to each cluster to capture the underlying shared time series patterns within the clusters. In this way, we use cluster identity to replace channel identity. Each Feed Forward is parameterized with a single linear layer due to its efficacy in time series forecasting [8, 15, 7]. Let $h_{\theta_k}(\cdot)$ represent the linear layer for the $k$-th cluster with weights $\theta_k$. $Z_i$ represents the hidden embedding of the $i$-th channel before the last layer. The final forecast is thereby averaged across the outputs of all cluster-aware Feed Forward with $\{p_{i,k}\}$ as weights, *e.g.*, $Y_i = \sum_k p_{i,k} h_{\theta_k}(Z_i)$ for the $i$-th channel. For computational efficiency, it is equivalent to $Y_i = h_{\theta^i}(Z_i)$ with averaged weights $\theta^i = \sum_k p_{i,k}\theta_k$.

**Univariate Adaptation**. In the context of univariate time series forecasting, we extend the proposed method to clustering on samples. We leverage the similarity between two univariate time series as defined in Eq. 1, and classify univariate time series with comparable patterns into the same cluster. This univariate adaptation allows it to capture interrelation within samples and extract valuable insights from analogous time series. This becomes particularly valuable in situations where meaningful dependencies exist among various univariate samples, such as the stock market.

**Zero-shot Forecasting**. Zero-shot forecasting is useful in time series applications where data privacy concerns restrict the feasibility of training models from scratch for unseen samples. The prototype embeddings acquired during the training phase serve as a compact representation of the pre-trained knowledge and can be harnessed for seamless knowledge transfer to unseen samples or new channels in a zero-shot setting. The pre-trained knowledge is applied to unseen instances by computing the clustering probability distribution on the pre-trained clusters, following Eq. 2, which is subsequently used for averaging cluster-aware Feed Forward. The cross-attention is disabled to fix the prototype embeddings in zero-shot forecasting. It is worth noting that zero-shot forecasting is applicable to both univariate and multivariate scenarios. We refer to Appendix B for detailed discussion.

## 4.3 Complexity Analysis

CCM effectively strikes a balance between the CI and CD strategies. On originally CI models, CCM introduces strategic clustering on channels, which not only reduces the model complexity but also enhances their generalizability. Simultaneously, CCM increases the model complexity on originally CD models with negligible overhead for higher capacities. We refer to Figure 5 for empirical analysis. Theoretically, the computational complexity of clustering probability computation (Eq. 2) and the cross-attention (Eq. 3) are $\mathcal{O}(KCd)$, where $K, C$ are the number of clusters and channels, respectively, and $d$ is the hidden dimension. One may also use other attention mechanisms [68, 69, 70] for efficiency. The complexity of cluster-aware Feed Forward scales linearly in $C, K$, and the forecasting horizon $H$.

# 5 Experiments

CCM consistently improves performance based on CI or CD models by significant margins across multiple benchmarks and settings, including long-term forecasting on 9 public multivariate datasets (Sec. 5.2); short-term forecasting on 2 univariate datasets (Sec. 5.3); and zero-shot forecasting in cross-domain and cross-granularity scenarios (Sec. 5.4).

## 5.1 Experimental Setup

**Datasets**. For long-term forecasting, we experiment on 9 popular benchmarking datasets across diverse domains [11, 12, 71], including weather, traffic and electricity. M4 dataset [72] is used in short-term forecasting, which is a univariate dataset that covers time series across diverse domains and various sampling frequencies from hourly to yearly. We further provide a new stock time series dataset with 1390 univariate time series. Each time series records the price history of an individual stock spanning 10 years. Due to the potential significant fluctuations in stock performance across different companies, this dataset poses challenges for capturing diverse and evolving stock patterns in financial markets. The statistics of long- and short-term datasets are shown in Table 2 and Table 3.

Table 2: The statistics of datasets in long-term forecasting. Horizon is $\{96, 192, 336, 720\}$.

| Dataset | Channels | Length | Frequency |
|---|---|---|---|
| ETTh1&ETTh2 | 7 | 17420 | 1 hour |
| ETTm1&ETTm2 | 7 | 69680 | 15 min |
| ILI | 7 | 966 | 1 week |
| Exchange | 8 | 7588 | 1 day |
| Weather | 21 | 52696 | 10 min |
| Electricity | 321 | 26304 | 1 hour |
| Traffic | 862 | 17544 | 1 hour |

Table 3: Dataset details of M4 and Stock in short-term forecasting.

| Dataset | Length | Horizon |
|---|---|---|
| M4 Yearly | 23000 | 6 |
| M4 Quarterly | 24000 | 8 |
| M4 Monthly | 48000 | 18 |
| M4 Weekly | 359 | 13 |
| M4 Daily | 4227 | 14 |
| M4 Hourly | 414 | 48 |
| **Stock (New)** | 10000 | 7/24 |

We follow standard protocols [11, 12, 13] for data splitting on public benchmarking datasets. As for the stock dataset, we divide the set of stocks into train/validation/test sets with a ratio of 7:2:1. Therefore, validation/test sets present unseen samples (*i.e.,* stocks) for model evaluation. This evaluation setting emphasizes the data efficiency aspect of time series models for scenarios where historical data is limited or insufficient for retraining from scratch given unseen instances. More details on datasets are provided in Appendix C.1.

**Base Models and Experimental Details**. CCM is a model-agnostic channel strategy that can be applied to arbitrary time series forecasting models for improved performance. We meticulously select four recent state-of-the-art time series models as base models: TSMixer [7], DLinear [8], PatchTST [21] and TimesNet [13], which mainly cover three mainstream paradigms, including linear models, transformer-based and convolutional models. For fair evaluation, we use the optimal experiment configuration as provided in the official code to implement both base models and the enhanced version with CCM. All the experiments are implemented with PyTorch on a single NVIDIA RTX A6000 48GB GPU. Experiment configurations and implementations are detailed in Appendix C.3. Experimental results in the following sections are averaged on five runs with different random seeds. Refer to Appendix C.6 for standard deviation results.

## 5.2 Long-term Forecasting Results

We report the mean squared error (MSE) and mean absolute error (MAE) on nine real-world datasets for long-term forecasting evaluation in Table 2. The forecasting horizon is $\{96, 192, 336, 720\}$. From the table, we observe that the model enhanced with CCM outperforms the base model in general. Specifically, CCM improves long-term forecasting performance in $90.27\%$ cases in MSE and $84.03\%$ cases in MAE across 144 different experiment settings. Remarkably, CCM achieves a substantial boost on DLinear, with a significant reduction on MSE by $5.12\%$ and MAE by $3.04\%$. The last column of the table quantifies the average percentage improvement in terms of MSE/MAE, which underscores the consistent enhancement brought by CCM across all forecasting horizons and datasets. Intuitively, the CCM method is more useful in scenarios where channel interactions are complex and significant, which is usually the case in real-world data. See more analysis in Appendix C.5.

Table 4: Long-term forecasting results on 9 real-world datasets in terms of MSE and MAE, the lower the better. The forecasting horizons are {96, 192, 336, 720}. The better performance in each setting is shown in **bold**. The best results for each row are underlined. The last column shows the average percentage of MSE/MAE improvement of CCM over four base models.

| Model / Metric | | TSMixer MSE | TSMixer MAE | + CCM MSE | + CCM MAE | DLinear MSE | DLinear MAE | + CCM MSE | + CCM MAE | PatchTST MSE | PatchTST MAE | + CCM MSE | + CCM MAE | TimesNet MSE | TimesNet MAE | + CCM MSE | + CCM MAE | IMP(%) |
|---|---|---|---|---|---|---|---|---|---|---|---|---|---|---|---|---|---|---|
| ETTh1 | 96 | 0.361 | 0.392 | 0.365 | 0.393 | 0.375 | 0.399 | 0.371 | 0.393 | 0.375 | 0.398 | 0.371 | 0.396 | 0.384 | 0.402 | 0.380 | 0.400 | 0.539 |
| | 192 | 0.404 | 0.418 | 0.402 | 0.418 | 0.405 | 0.416 | 0.404 | 0.415 | 0.415 | 0.425 | 0.414 | 0.424 | 0.436 | 0.429 | 0.431 | 0.425 | 0.442 |
| | 336 | 0.422 | 0.430 | 0.423 | 0.430 | 0.445 | 0.440 | 0.438 | 0.443 | 0.422 | 0.440 | 0.417 | 0.429 | 0.491 | 0.469 | 0.485 | 0.461 | 0.908 |
| | 720 | 0.463 | 0.472 | 0.462 | 0.470 | 0.489 | 0.488 | 0.479 | 0.497 | 0.449 | 0.468 | 0.447 | 0.469 | 0.521 | 0.500 | 0.520 | 0.493 | 0.333 |
| ETTm1 | 96 | 0.285 | 0.339 | 0.283 | 0.337 | 0.299 | 0.343 | 0.298 | 0.343 | 0.294 | 0.351 | 0.289 | 0.338 | 0.338 | 0.375 | 0.335 | 0.371 | 1.123 |
| | 192 | 0.339 | 0.365 | 0.336 | 0.368 | 0.335 | 0.365 | 0.334 | 0.365 | 0.334 | 0.370 | 0.333 | 0.363 | 0.374 | 0.387 | 0.373 | 0.383 | 0.482 |
| | 336 | 0.361 | 0.406 | 0.359 | 0.393 | 0.370 | 0.386 | 0.365 | 0.385 | 0.373 | 0.397 | 0.370 | 0.392 | 0.410 | 0.411 | 0.412 | 0.416 | 0.716 |
| | 720 | 0.445 | 0.470 | 0.424 | 0.421 | 0.427 | 0.423 | 0.424 | 0.417 | 0.416 | 0.420 | 0.419 | 0.430 | 0.478 | 0.450 | 0.477 | 0.448 | 1.852 |
| ETTh2 | 96 | 0.284 | 0.343 | 0.278 | 0.338 | 0.289 | 0.353 | 0.285 | 0.348 | 0.278 | 0.340 | 0.274 | 0.336 | 0.340 | 0.374 | 0.336 | 0.371 | 1.371 |
| | 192 | 0.339 | 0.385 | 0.325 | 0.393 | 0.384 | 0.418 | 0.376 | 0.413 | 0.341 | 0.382 | 0.339 | 0.355 | 0.402 | 0.414 | 0.400 | 0.410 | 1.806 |
| | 336 | 0.361 | 0.406 | 0.361 | 0.399 | 0.442 | 0.459 | 0.438 | 0.455 | 0.329 | 0.384 | 0.327 | 0.383 | 0.452 | 0.452 | 0.449 | 0.445 | 0.823 |
| | 720 | 0.445 | 0.470 | 0.438 | 0.464 | 0.601 | 0.549 | 0.499 | 0.496 | 0.381 | 0.424 | 0.378 | 0.415 | 0.462 | 0.468 | 0.457 | 0.461 | 4.370 |
| ETTm2 | 96 | 0.171 | 0.260 | 0.167 | 0.260 | 0.167 | 0.260 | 0.166 | 0.258 | 0.174 | 0.261 | 0.168 | 0.256 | 0.187 | 0.267 | 0.189 | 0.270 | 0.860 |
| | 192 | 0.221 | 0.296 | 0.220 | 0.296 | 0.284 | 0.352 | 0.243 | 0.323 | 0.238 | 0.307 | 0.231 | 0.300 | 0.249 | 0.309 | 0.250 | 0.310 | 3.453 |
| | 336 | 0.276 | 0.329 | 0.277 | 0.330 | 0.369 | 0.427 | 0.295 | 0.358 | 0.293 | 0.346 | 0.275 | 0.331 | 0.321 | 0.351 | 0.318 | 0.347 | 6.012 |
| | 720 | 0.420 | 0.422 | 0.369 | 0.391 | 0.554 | 0.522 | 0.451 | 0.456 | 0.373 | 0.401 | 0.374 | 0.400 | 0.408 | 0.403 | 0.394 | 0.391 | 7.139 |
| Exchange | 96 | 0.089 | 0.209 | 0.085 | 0.206 | 0.088 | 0.215 | 0.085 | 0.214 | 0.094 | 0.216 | 0.088 | 0.208 | 0.107 | 0.234 | 0.105 | 0.231 | 2.880 |
| | 192 | 0.195 | 0.315 | 0.177 | 0.300 | 0.178 | 0.317 | 0.171 | 0.306 | 0.191 | 0.311 | 0.185 | 0.309 | 0.226 | 0.344 | 0.224 | 0.340 | 3.403 |
| | 336 | 0.343 | 0.421 | 0.312 | 0.405 | 0.371 | 0.462 | 0.300 | 0.300 | 0.343 | 0.427 | 0.342 | 0.423 | 0.367 | 0.448 | 0.361 | 0.442 | 5.875 |
| | 720 | 0.898 | 0.710 | 0.847 | 0.697 | 0.966 | 0.754 | 0.811 | 0.683 | 0.888 | 0.706 | 0.813 | 0.673 | 0.964 | 0.746 | 0.957 | 0.739 | 5.970 |
| ILI | 24 | 1.914 | 0.879 | 1.938 | 0.874 | 2.215 | 1.081 | 1.935 | 0.935 | 1.593 | 0.757 | 1.561 | 0.750 | 2.317 | 0.934 | 2.139 | 0.936 | 4.483 |
| | 36 | 1.808 | 0.858 | 1.800 | 0.851 | 2.142 | 0.977 | 1.938 | 0.942 | 1.768 | 0.794 | 1.706 | 0.780 | 1.972 | 0.920 | 1.968 | 0.914 | 2.561 |
| | 48 | 1.797 | 0.873 | 1.796 | 0.867 | 2.335 | 1.056 | 2.221 | 1.030 | 1.799 | 0.916 | 1.774 | 0.892 | 2.238 | 0.940 | 2.229 | 0.937 | 1.602 |
| | 60 | 1.859 | 0.895 | 1.810 | 0.876 | 2.479 | 1.088 | 2.382 | 1.096 | 1.850 | 0.943 | 1.735 | 0.880 | 2.027 | 0.928 | 2.041 | 0.930 | 2.491 |
| Weather | 96 | 0.149 | 0.198 | 0.147 | 0.194 | 0.192 | 0.250 | 0.187 | 0.245 | 0.149 | 0.198 | 0.147 | 0.197 | 0.172 | 0.220 | 0.169 | 0.215 | 1.729 |
| | 192 | 0.201 | 0.248 | 0.192 | 0.242 | 0.248 | 0.297 | 0.240 | 0.285 | 0.194 | 0.241 | 0.191 | 0.238 | 0.219 | 0.261 | 0.215 | 0.257 | 2.539 |
| | 336 | 0.264 | 0.291 | 0.244 | 0.281 | 0.284 | 0.335 | 0.274 | 0.324 | 0.244 | 0.282 | 0.245 | 0.285 | 0.280 | 0.306 | 0.274 | 0.291 | 2.924 |
| | 720 | 0.320 | 0.336 | 0.318 | 0.334 | 0.339 | 0.374 | 0.320 | 0.357 | 0.320 | 0.334 | 0.316 | 0.333 | 0.365 | 0.359 | 0.366 | 0.362 | 1.476 |
| Electricity | 96 | 0.142 | 0.237 | 0.139 | 0.235 | 0.153 | 0.239 | 0.142 | 0.247 | 0.138 | 0.233 | 0.136 | 0.231 | 0.168 | 0.272 | 0.158 | 0.259 | 2.480 |
| | 192 | 0.154 | 0.248 | 0.147 | 0.246 | 0.158 | 0.251 | 0.152 | 0.248 | 0.153 | 0.247 | 0.153 | 0.248 | 0.184 | 0.289 | 0.172 | 0.262 | 3.226 |
| | 336 | 0.163 | 0.264 | 0.161 | 0.262 | 0.170 | 0.269 | 0.168 | 0.267 | 0.170 | 0.263 | 0.168 | 0.262 | 0.198 | 0.300 | 0.181 | 0.284 | 2.423 |
| | 720 | 0.208 | 0.300 | 0.204 | 0.299 | 0.233 | 0.342 | 0.230 | 0.338 | 0.206 | 0.296 | 0.210 | 0.301 | 0.220 | 0.320 | 0.205 | 0.309 | 1.417 |
| Traffic | 96 | 0.376 | 0.264 | 0.375 | 0.262 | 0.411 | 0.284 | 0.411 | 0.282 | 0.360 | 0.249 | 0.357 | 0.246 | 0.593 | 0.321 | 0.554 | 0.316 | 1.488 |
| | 192 | 0.397 | 0.277 | 0.340 | 0.279 | 0.423 | 0.287 | 0.422 | 0.286 | 0.379 | 0.256 | 0.379 | 0.254 | 0.617 | 0.336 | 0.562 | 0.331 | 3.175 |
| | 336 | 0.413 | 0.290 | 0.411 | 0.289 | 0.438 | 0.299 | 0.436 | 0.297 | 0.401 | 0.270 | 0.389 | 0.255 | 0.629 | 0.336 | 0.579 | 0.341 | 2.120 |
| | 720 | 0.444 | 0.306 | 0.441 | 0.302 | 0.467 | 0.316 | 0.471 | 0.318 | 0.443 | 0.294 | 0.430 | 0.281 | 0.640 | 0.350 | 0.587 | 0.366 | 1.445 |

**Comparison with Regularization Solution**. PRReg [17] recently displays to improve the performance of CD models, by predicting residuals with a regularization term in the training objective. We evaluate the effectiveness of PRReg and our proposed CCM on long-term forecasting performance enhancement based on CI and CD models. Following the previous training setting [17], we develop CI and CD versions for Linear [8] and Transformer [41] and report MSE loss. Table 5 shows that CCM surpasses PRReg in most cases, highlighting its efficacy compared with regularization solutions. See full results in Appendix C.4.

## 5.3 Short-term Forecasting Results

In both M4 and stock datasets, we follow the univariate forecasting setting. For M4 benchmarks, we adopt the evaluation setting in prior works [42] and report the symmetric mean absolute percentage error (SMAPE), mean absolute scaled error (MASE), and overall weighted average (OWA). As for the stock dataset, we implement MAE and MSE as metrics in Table 6. See more details on metrics in Appendix C.2. Remarkably, the efficacy of CCM is consistent across all M4 sub-datasets with different sampling frequencies. Specifically, CCM outperforms the state-of-the-art linear model (DLinear) by a significant margin of 11.62%, and outperforms the best convolutional method TimesNet by 8.88%. In Table 6, we also observe a significant performance improvement on the stock dataset, achieved by applying CCM. In the stock

Table 5: Comparison between CCM and existing regularization method for improved performance on CI/CD strategies. The best results are highlighted in **bold**. The forecasting horizon is 24 for ILI dataset and 48 for other datasets. ⋆ denotes our implementation. Other results collect from [17]

| | | CD | CI | +PRReg | +CCM⋆ |
|---|---|---|---|---|---|
| ETTh1 | Linear | 0.402 | 0.345 | **0.342** | **0.342** |
| | Transformer | 0.861 | 0.655 | 0.539 | **0.518** |
| ETTm1 | Linear | 0.404 | 0.354 | 0.311 | **0.310** |
| | Transformer | 0.458 | 0.379 | 0.349 | **0.300** |
| Weather | Linear | 0.142 | 0.169 | 0.131 | **0.130** |
| | Transformer | 0.251 | 0.168 | 0.180 | **0.164** |
| ILI | Linear | 2.343 | 2.847 | 2.299 | **2.279** |
| | Transformer | 5.309 | 4.307 | 3.254 | **3.206** |
| Electricity | Linear | **0.195** | 0.196 | 0.196 | **0.195** |
| | Transformer | 0.250 | 0.185 | 0.185 | **0.183** |

dataset, we test on new samples (*i.e.,* univariate stock time series) that the model has not seen during

Table 6: Short-term forecasting results on M4 dataset in terms of SMAPE, MASE, and OWA, and stock dataset in terms of MSE and MAE. The lower the better. The forecasting horizon is $\{7, 24\}$ for the stock dataset. The better performance in each setting is shown in **bold**.

| Model | | TSMixer | + CCM | DLinear | + CCM | PatchTST | + CCM | TimesNet | + CCM | IMP(%) |
|---|---|---|---|---|---|---|---|---|---|---|
| M4 (Yearly) | SMAPE | 14.702 | **14.676** | 16.965 | **14.337** | 13.477 | **13.304** | 15.378 | **14.426** | 7.286 |
| | MASE | **3.343** | 3.370 | 4.283 | **3.144** | 3.019 | **2.997** | 3.554 | **3.448** | 9.589 |
| | OWA | 0.875 | **0.873** | 1.058 | **0.834** | 0.792 | **0.781** | 0.918 | **0.802** | 11.346 |
| M4 (Quarterly) | SMAPE | 11.187 | **10.989** | 12.145 | **10.513** | 10.380 | **10.359** | 10.465 | **10.121** | 6.165 |
| | MASE | 1.346 | **1.332** | 1.520 | **1.243** | 1.233 | **1.224** | 1.227 | **1.183** | 7.617 |
| | OWA | 0.998 | **0.984** | 1.106 | **0.931** | 0.921 | **0.915** | 0.923 | **0.897** | 6.681 |
| M4 (Monthly) | SMAPE | 13.433 | **13.407** | 13.514 | **13.370** | 12.959 | **12.672** | 13.513 | **12.790** | 2.203 |
| | MASE | 1.022 | **1.019** | 1.037 | **1.005** | 0.970 | **0.941** | 1.039 | **0.942** | 4.238 |
| | OWA | 0.946 | **0.944** | 0.956 | **0.936** | 0.905 | **0.895** | 0.957 | **0.891** | 3.067 |
| M4 (Others) | SMAPE | **7.067** | 7.178 | 6.709 | **6.160** | 4.952 | **4.643** | 6.913 | **5.218** | 10.377 |
| | MASE | 5.587 | **5.302** | 4.953 | **4.713** | 3.347 | **3.128** | 4.507 | **3.892** | 7.864 |
| | OWA | 1.642 | **1.536** | 1.487 | **1.389** | 1.049 | **0.997** | 1.438 | **1.217** | 9.472 |
| M4 (Avg.) | SMAPE | 12.867 | **12.807** | 13.639 | **12.546** | 12.059 | **11.851** | 12.880 | **11.914** | 5.327 |
| | MASE | 1.887 | **1.864** | 2.095 | **1.740** | 1.623 | **1.587** | 1.836 | **1.603** | 10.285 |
| | OWA | 0.957 | **0.948** | 1.051 | **0.917** | 0.869 | **0.840** | 0.955 | **0.894** | 6.693 |
| Stock (Horizon 7) | MSE | 0.939 | **0.938** | 0.992 | **0.883** | 0.896 | **0.892** | 0.930 | **0.915** | 3.288 |
| | MAE | 0.807 | **0.806** | 0.831 | **0.774** | 0.771 | **0.771** | 0.802 | **0.793** | 2.026 |
| Stock (Horizon 24) | MSE | 1.007 | **0.991** | 0.996 | **0.917** | 0.930 | **0.880** | 0.998 | **0.937** | 5.252 |
| | MAE | 0.829 | **0.817** | 0.832 | **0.781** | 0.789 | **0.765** | 0.830 | **0.789** | 3.889 |

training to evaluate the model's generalization and robustness. By memorizing cluster-specific knowledge from analogous samples, the model potentially captures various market trends and behaviors and thereby makes more accurate and informed forecasting.

Table 7: Zero-shot forecasting results on ETT datasets. The forecasting horizon is $\{96, 720\}$. The best value in each row is underlined.

| Model Generalization Task | | TSMixer | | + CCM | | DLinear | | + CCM | | PatchTST | | + CCM | | TimesNet | | + CCM | | IMP(%) |
|---|---|---|---|---|---|---|---|---|---|---|---|---|---|---|---|---|---|---|
| | | MSE | MAE | MSE | MAE | MSE | MAE | MSE | MAE | MSE | MAE | MSE | MAE | MSE | MAE | MSE | MAE | |
| ① ETTh1→ETTh2 | 96 | 0.288 | 0.357 | 0.283 | 0.353 | 0.308 | 0.371 | 0.283 | 0.349 | 0.313 | 0.362 | 0.292 | 0.346 | 0.391 | 0.412 | 0.388 | 0.410 | 3.661 |
| | 720 | 0.374 | 0.414 | 0.370 | 0.413 | 0.569 | 0.549 | 0.520 | 0.517 | 0.414 | 0.442 | 0.386 | 0.423 | 0.540 | 0.508 | 0.516 | 0.491 | 4.326 |
| ② ETTh1→ETTm1 | 96 | 0.763 | 0.677 | 0.710 | 0.652 | 0.726 | 0.658 | 0.681 | 0.634 | 0.729 | 0.667 | 0.698 | 0.647 | 0.887 | 0.718 | 0.827 | 0.700 | 4.626 |
| | 720 | 1.252 | 0.815 | 1.215 | 0.803 | 1.881 | 0.948 | 1.138 | 0.809 | 1.459 | 0.845 | 1.249 | 0.795 | 1.623 | 0.981 | 1.601 | 0.964 | 10.249 |
| ③ ETTh1→ETTm2 | 96 | 0.959 | 0.694 | 0.937 | 0.689 | 0.990 | 0.704 | 0.896 | 0.677 | 0.918 | 0.694 | 0.895 | 0.677 | 1.199 | 0.794 | 1.122 | 0.731 | 4.457 |
| | 720 | 1.765 | 0.982 | 1.758 | 0.980 | 2.091 | 1.061 | 1.681 | 0.954 | 1.925 | 1.014 | 1.718 | 0.966 | 2.204 | 1.031 | 1.874 | 1.012 | 7.824 |
| ④ ETTh2→ETTh1 | 96 | 0.466 | 0.462 | 0.455 | 0.456 | 0.462 | 0.450 | 0.427 | 0.432 | 0.620 | 0.563 | 0.509 | 0.495 | 0.869 | 0.624 | 0.752 | 0.590 | 8.016 |
| | 720 | 0.695 | 0.584 | 0.540 | 0.519 | 0.511 | 0.518 | 0.484 | 0.502 | 1.010 | 0.968 | 0.936 | 0.686 | 1.274 | 0.783 | 0.845 | 0.642 | 16.243 |
| ⑤ ETTh2→ETTm2 | 96 | 0.943 | 0.726 | 0.876 | 0.697 | 0.736 | 0.656 | 0.700 | 0.642 | 0.840 | 0.708 | 0.771 | 0.688 | 1.250 | 0.850 | 1.064 | 0.793 | 6.344 |
| | 720 | 1.472 | 0.872 | 1.464 | 0.866 | 1.813 | 0.938 | 1.253 | 0.844 | 1.832 | 1.052 | 1.532 | 0.863 | 1.861 | 1.016 | 1.671 | 0.967 | 11.439 |
| ⑥ ETTh2→ETTm1 | 96 | 1.254 | 0.771 | 1.073 | 0.714 | 1.147 | 0.746 | 0.894 | 0.669 | 0.997 | 0.721 | 0.789 | 0.629 | 1.049 | 0.791 | 0.804 | 0.657 | 16.016 |
| | 720 | 2.275 | 1.137 | 1.754 | 1.065 | 1.992 | 1.001 | 1.740 | 0.970 | 2.651 | 1.149 | 1.695 | 0.971 | 2.183 | 1.103 | 1.742 | 0.983 | 15.952 |

## 5.4 Zero-shot Forecasting Results

Existing time series models tend to be rigidly tailored to a specific dataset, leading to poor generalization on unseen data. In contrast, CCM leverages learned prototypes to capture cluster-specific knowledge. This enables meaningful comparisons between unseen time series and pre-trained knowledge, facilitating accurate zero-shot forecasting. Following prior work [73], we adopt ETT collections [11], where ETTh1 and ETTh2 are hourly recorded, while ETTm1 and ETTm2 are minutely recorded. "1" and "2" indicate two different regions where the datasets originated. Table 7 shows MSE and MAE results on test datasets. CCM consistently improves the zero-shot forecasting capacity of base time series models in 48 scenarios, including generalization to different regions and different granularities. Specifically, based on the results, we make the following observations. (1) CCM exhibits more significant performance improvement with longer forecasting horizons, highlighting the efficacy of memorizing and leveraging pre-trained knowledge in zero-shot forecasting scenarios. (2) CCM demonstrates a better effect on originally CI base models. For instance, the averaged improvement rates on two CI models, *i.e.,* DLinear and PatchTST, are 10.48% and 11.13% respectively, while the improvement rates on TSMixer and TimesNet are 5.14% and 9.63%. Overall, the experimental results verify the superiority and efficacy of CCM in enhancing zero-shot forecasting and the practical value of generalization within closely related domains under varying conditions.

## 5.5 Qualitative Visualization

**Channel Clustering Visualization**. Figure 2 presents the t-SNE visualization of channel and prototype embeddings within ETTh1 and ETTh2 datasets with DLinear as the base model. Each point represents a channel within a sample, with varying colors indicating different channels. In ETTh1 dataset, we discern a pronounced clustering of channels 0, 2, and 4, suggesting that they may

be capturing related or redundant information within the dataset. Concurrently, channels 1, 3, 5, and 6 coalesce into another cluster.

The similarity matrix in the lower left further corroborates these findings. Clustering is also observable in ETTh2 dataset, particularly among channels 0, 4, and 5, as well as channels 2, 3, and 6. Comparatively, channel 1 shows a dispersion among clusters, partly due to its capturing of unique or diverse aspects of the data that do not closely align with the features represented by any clusters. The clustering results demonstrate that CCM not only elucidates the intricate relationships and potential redundancies among the channels but also offers critical insights for feature analysis and enhancing the interpretability of time series models.

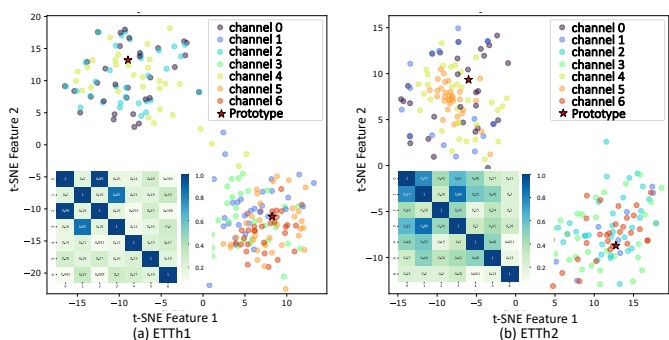

Figure 2: t-SNE visualization of channel and prototype embedding by DLinear with CCM on (a) ETTh1 and (b) ETTh2 dataset. The lower left corner shows the similarity matrix between channels.

**Weight Visualization of Cluster-aware Projection**. Figure 3 depicts the weights visualization for the cluster-aware Feed Forward on ETTh1 and ETTm1 datasets, revealing distinct patterns that are indicative of the model's learned features [15, 8, 51].

For instance, in the ETTm1 dataset, Cluster 0 shows bright diagonal striping patterns, which may suggest that it is primarily responsible for capturing the most dominant periodic signals in the corresponding cluster. In contrast, Cluster 1 exhibits denser stripes, indicating its role in refining the representation by capturing more subtle or complex periodicities that the first layer does not. The visualization implies the model's ability to identify and represent periodicity in diverse patterns, which is crucial for time-series forecasting tasks that are characterized by intricate cyclic behaviors.

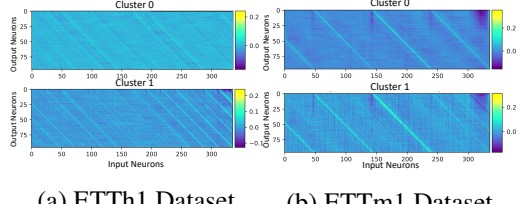

(a) ETTh1 Dataset      (b) ETTm1 Dataset

Figure 3: Weights visualization of cluster-wise linear layers on (a) ETTh1 and (b) ETTm1 datasets. The input and output lengths are 336 and 96, respectively. We observe the different periodicities captured by different clusters.

### 5.6 Ablation Studies

Figure 4 shows an ablation study on cluster ratios, which is defined as the ratio of the number of clusters to the number of channels. 0.0 means all channels are in a single cluster. We observe that the MSE loss slightly decreases and then increases as the cluster ratio increases, especially for DLinear, PatchTST, and TimesNet. Time series models with CCM achieve the best performance when the cluster ratio is in the range of $[0.2, 0.6]$. It is worth noticing that DLinear and PatchTST, two CI models among four base models, benefit consistently from channel clustering with any number of clusters. Additional ablation studies on the look-back window length and clustering step are provided in Appendix D.

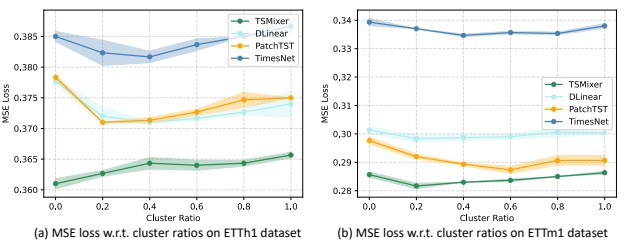

Figure 4: Ablation Study on Cluster Ratios in terms of MSE loss with four base models. The forecasting horizon is 96. (*left*: ETTh1 dataset; *right*: ETTm1 dataset)

## 5.7 Efficiency Analysis

We evaluate the model size and runtime efficiency of the proposed CCM with various numbers of clusters on ETTh1 dataset, as shown in Figure 5. The batch size is 32, and the hidden dimension is 64. We keep all other hyperparameters consistent to ensure fair evaluation. It is worth noting that CCM reduces the model complexity based on Channel-Independent models (*e.g.,* PatchTST, DLinear), since CCM essentially uses cluster identity to replace channel identity. The generalizability of CI models is thereby enhanced as well. When it comes to Channel-Dependent models, CCM increases the model complexity with negligible overhead, considering the improved forecasting performance.

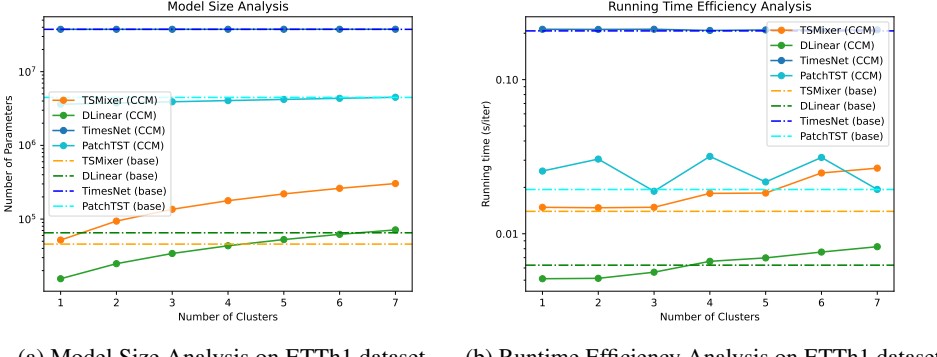

(a) Model Size Analysis on ETTh1 dataset      (b) Runtime Efficiency Analysis on ETTh1 dataset

Figure 5: Efficiency analysis in model size and running time on ETTh1 dataset.

## 6 Conclusion

This work introduces a novel Channel Clustering Module (CCM) to address the challenge of effective channel management in time series forecasting. CCM strikes a balance between individual channel treatment and capturing cross-channel dependencies by clustering channels based on their intrinsic similarity. Extensive experiments demonstrate the efficacy of CCM in multiple benchmarks, including long-term, short-term, and zero-shot forecasting scenarios. Refinement of the CCM clustering and domain-specific similarity measurement could potentially improve the model performance further. Moreover, it would be valuable to investigate the applicability of CCM in other domains beyond time series forecasting in future works.

## Acknowledgments and Disclosure of Funding

This research was supported in part by the National Science Foundation (NSF) CNS Division Of Computer and Network Systems (2431504), NSF-AoF FAIN (2132573), ARO (W911NF-23-1-0088) and AWS Research Awards. We would like to thank the anonymous reviewers for their constructive feedback. Their contributions have been invaluable in facilitating our work.

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

# A Definitions

## A.1 Channel Similarity

Essentially, the similarity between two time series $X_i$ and $X_j$ is defined as $\text{SIM}(X_i, X_j) = \exp(\frac{-d(X_i, X_j)}{2\sigma^2})$, where $d(\cdot, \cdot)$ can be any distance metric [74, 75], such as Euclidean Distance ($L_p$), Editing Distance (ED) and Dynamic Time Warping (DTW) [76]. One may also use other similarity definitions, such as Longest Common Subsequence (LCSS) and Cross-correlation (CCor).

Firstly, the choice of Euclidean distance in this work is motivated by its efficiency and low computational complexity, especially in the case of large datasets or real-time applications. Let $H$ denote the length of the time series. The complexity of the above similarity computation is shown in Table 8.

Secondly, it's worth noting that while there are various similarity computation approaches, studies have demonstrated a strong correlation between Euclidean distance and other distance metrics [77]. This high correlation suggests that, despite different mathematical formulations, these metrics often yield similar results

Table 8: Complexity of similarity computation

| Euclidean | Edit Distance | DTW | LCSS | CCor |
|-----------|---------------|-----|------|------|
| $\mathcal{O}(H)$ | $\mathcal{O}(H^2)$ | $\mathcal{O}(H^2)$ | $\mathcal{O}(H^2)$ | $\mathcal{O}(H^2)$ |

when assessing the similarity between time series. This empirical evidence supports the choice of Euclidean distance as a reasonable approximation of similarity for practical purposes. In our implementation, we select $\sigma = 5$ in Eq. 1 to compute the similarities based on Euclidean distance.

## A.2 Channel Dependent and Channel Independent Strategy

The definitions of Channel Dependent (CD) and Channel Independent (CI) settings are pivotal to this work. The fundamental difference lies in whether a model captures cross-channel information. There are slightly varied interpretations of Channel Independent (CI) in previous works and we summarize as follows.

1. In some works [15, 54], CI is broadly defined as forecasting each channel independently, where cross-channel dependencies are completely ignored. For linear models [78, 16], CI is specifically defined as $n$ individual linear layers for $n$ channels in previous works. Each linear layer is dedicated to modeling a univariate sequence, with the possibility of differing linear weights across channels.

2. In previous work [17], CI also means all channels being modeled independently yet under a unified model.

All the above works acknowledge that CI strategies often outperform CD approaches, though this comparison is not the focal point of our work. It's also recognized that the specific CI strategy employed in DLinear and PatchTST contributes significantly to their performance. The CI setting in [17] represents a specific instance within the broader CI setting in other works [15, 54]. To avoid ambiguity, we use $f^{(i)}$ to represent the model for the $i$-th channel specifically, aligning with previous definitions without conflict.

# B Multivariate and Univariate Adaptation

We provide pseudocodes for training time series models enhanced with CCM in Algorithm 1. Algorithm 2 displays pseudocodes for the inference phase, where both the training and test sets have the same number of channels. The components in the pretrained model $\mathcal{F}$, highlighted in blue, remain fixed during the inference phase. It's important to note that zero-shot forecasting in Algorithm 2 is adaptable to various scenarios. Let's discuss these scenarios:

- **Training on a univariate dataset and inferring on either univariate or multivariate samples:** In this case, the model learns prototypes from a vast collection of univariate time series in the training set. As a result, the model can effortlessly adapt to forecasting unseen univariate time series in a zero-shot manner. To forecast unseen multivariate time series, we decompose each multivariate sample into multiple univariate samples, where each univariate sample can be processed by the pretrained model. The future multivariate time series can be obtained by stacking multiple future univariate time series.

- **Training on a multivariate dataset and inferring on either univariate or multivariate samples:** For Channel-Dependent models, test samples should have the same number of channels as the training samples, as seen in sub-datasets within ETT collections [11]. In contrast, for Channel-Independent models, zero-shot forecasting can be performed on either univariate or multivariate samples, even when they have different numbers of channels.

---

**Algorithm 1** Forward function of time series models with channel clustering module. $C$ is the number of channels in the dataset. $K$ is the number of clusters. $T$ is the length of historical data. $H$ is the forecasting horizon.

---

**Input:** Historical time series $X \in \mathbb{R}^{T \times C}$
**Output:** Future time series $Y \in \mathbb{R}^{H \times C}$; Prototype embedding $\mathbf{C} \in \mathbb{R}^{K \times d}$
Initialize the weights of $K$ linear layer $\theta_k$ for $k = 1, \cdots, K$
Initialize $K$ cluster embedding $c_k \in \mathbb{R}^d$ for $k = 1, \cdots, K$.      $\triangleright$ Cluster Embedding $\mathbf{C}$
$X \leftarrow \text{Normalize}(X)$
$\mathbf{S}_{i,j} \leftarrow \exp(\frac{-\|X_i - X_j\|^2}{2\sigma^2})$.      $\triangleright$ Compute Similarity Matrix $\mathbf{S}$
$h_i \leftarrow \text{MLP}(X_i)$ for each channel $i$.      $\triangleright$ Channel Embedding $\mathbf{H}$ via MLP Encoder in the Cluster Assigner
$p_{i,k} \leftarrow \text{Normalize}(\frac{c_k^\top h_i}{\|c_k\|\|h_i\|}) \in [0,1]$.      $\triangleright$ Compute Clustering Probability Matrix $\mathbf{P}$
$\mathbf{M} \leftarrow \text{Bernoulli}(\mathbf{P})$.      $\triangleright$ Sample Clustering Membership Matrix $\mathbf{M}$
$\mathbf{C} \leftarrow \text{Normalize}\left(\exp(\frac{(W_Q\mathbf{C})(W_K\mathbf{H})^\top}{\sqrt{d}}) \odot \mathbf{M}^\top\right) W_V \mathbf{H}$.      $\triangleright$ Update Cluster Embedding $\mathbf{C}$ via Cross Attention
$\hat{\mathbf{H}} = \text{Temporal Module}(\mathbf{H})$.      $\triangleright$ Update via Temporal Modules
**for** channel $i$ in $\{1, 2, \cdots, C\}$ **do**
     $Y_i \leftarrow h_{\theta^i}(\hat{\mathbf{H}}_i)$ where $\theta^i = \sum_k p_{i,k}\theta_k$.      $\triangleright$ Weight Averaging and Projection
**end for**

---

**Algorithm 2** Zero-shot forecasting of time series models with channel clustering module. $C$ is the number of channels in both the training and test datasets. $K$ is the number of clusters. $T$ is the length of historical data. $H$ is the forecasting horizon.

---

**Input:** Historical time series $X \in \mathbb{R}^{T \times C}$; Pretrained Models $\mathcal{F}$
**Output:** Future time series $Y \in \mathbb{R}^{H \times C}$;
Load the weights of $K$ linear layer $\theta_k$ for $k = 1, \cdots, K$ from $\mathcal{F}$
Load $K$ cluster embedding $c_k \in \mathbb{R}^d$ for $k = 1, \cdots, K$ from $\mathcal{F}$.      $\triangleright$ Cluster Embedding $\mathbf{C}$
$X \leftarrow \text{Normalize}(X)$
$\mathbf{S}_{i,j} \leftarrow \exp(\frac{-\|X_i - X_j\|^2}{2\sigma^2})$.      $\triangleright$ Compute Similarity Matrix $\mathbf{S}$
$h_i \leftarrow \text{MLP}(X_i)$ for each channel $i$.      $\triangleright$ Channel Embedding $\mathbf{H}$ via MLP Encoder in the Cluster Assigner
$p_{i,k} \leftarrow \text{Normalize}(\frac{c_k^\top h_i}{\|c_k\|\|h_i\|}) \in [0,1]$.      $\triangleright$ Compute Clustering Probability Matrix $\mathbf{P}$
$\mathbf{M} \leftarrow \text{Bernoulli}(\mathbf{P})$.      $\triangleright$ Sample Clustering Membership Matrix $\mathbf{M}$
$\hat{\mathbf{H}} = \text{Temporal Module}(\mathbf{H})$.      $\triangleright$ Update via Temporal Modules
**for** channel $i$ in $\{1, 2, \cdots, C\}$ **do**
     $Y_i \leftarrow h_{\theta^i}(\hat{\mathbf{H}}_i)$ where $\theta^i = \sum_k p_{i,k}\theta_k$.      $\triangleright$ Weight Averaging and Projection
**end for**

---

## C Experiments

### C.1 Datasets

**Public Datasets**. We utilize nine commonly used datasets for long-term forecasting evaluation. Firstly, ETT collection [11], which documents the oil temperature and load features of electricity transformers over the period spanning July 2016 to July 2018. This dataset is further subdivided into four sub-datasets, ETTh*s* and ETTm*s*, with hourly and 15-minute sampling frequencies, respectively.

$s$ can be 1 or 2, indicating two different regions. Electricity dataset [79] encompasses electricity consumption data from 321 clients from July 2016 to July 2019. Exchange [80] compiles daily exchange rate information from 1990 to 2016. Traffic dataset contains hourly traffic load data from 862 sensors in San Francisco areas from 2015 to 2016. Weather dataset offers a valuable resource with 21 distinct weather indicators, including air temperature and humidity, collected every 10 minutes throughout the year 2021. ILI documents the weekly ratio of influenza-like illness patients relative to the total number of patients, spanning from 2002 to 2021. Dataset statistics can be found in Table 9.

We adopt M4 dataset for short-term forecasting evaluation, which involves 100,000 univariate time series collected from different domains, including finance, industry, *etc.*The M4 dataset is further divided into 6 sub-datasets, according to the sampling frequency.

Table 9: The statistics of dataset in long-term and short-term forecasting tasks

| Tasks | Dataset | Channels | Forecast Horizon | Length | Frequency | Domain |
|---|---|---|---|---|---|---|
| Long-term | ETTh1 | 7 | $\{96, 192, 336, 720\}$ | 17420 | 1 hour | Temperature |
| | ETTh2 | 7 | $\{96, 192, 336, 720\}$ | 17420 | 1 hour | Temperature |
| | ETTm1 | 7 | $\{96, 192, 336, 720\}$ | 69680 | 15 min | Temperature |
| | ETTm2 | 7 | $\{96, 192, 336, 720\}$ | 69680 | 15 min | Temperature |
| | Illness | 7 | $\{96, 192, 336, 720\}$ | 966 | 1 week | Illness Ratio |
| | Exchange | 8 | $\{96, 192, 336, 720\}$ | 7588 | 1 day | Exchange Rates |
| | Weather | 21 | $\{96, 192, 336, 720\}$ | 52696 | 10 min | Weather |
| | Electricity | 321 | $\{96, 192, 336, 720\}$ | 26304 | 1 hour | Electricity |
| | Traffic | 862 | $\{96, 192, 336, 720\}$ | 17544 | 1 hour | Traffic Load |
| Short-term | M4-Yearly | 1 | 6 | 23000 | yearly | Demographic |
| | M4 Quarterly | 1 | 8 | 24000 | quarterly | Finance |
| | M4 Monthly | 1 | 18 | 48000 | monthly | Industry |
| | M4 Weekly | 1 | 13 | 359 | weekly | Macro |
| | M4 Daily | 1 | 14 | 4227 | daily | Micro |
| | M4 Hourly | 1 | 48 | 414 | hourly | Other |
| | **Stock** | 1 | $\{7, 24\}$ | 10000 | 5 min | Stock |

**Stock Dataset**. We design a new time series benchmark dataset constructed from publicly available stock-market data. We deploy commercial stock market API to probe the market statistics for 1390 stocks spanning 10 years from Nov.25, 2013 to Sep.1, 2023. We collect stock price data from 9:30 a.m. to 4:00 p.m. every stock open day except early closure days. The sampling granularity is set to be 5 minutes. Missing record is recovered by interpolation from nearby timestamps and all stock time series are processed to have aligned timestep sequences. We implement short-term forecasting on the stock dataset, which is more focused on market sentiment, and short-term events that can cause stock prices to fluctuate over days, weeks, or months. Thereby, we set the forecasting horizon as 7 and 24.

## C.2 Metrics

Following standard evaluation protocols [13], we utilize the Mean Absolute Error (MAE) and Mean Square Error(MSE) for long-term and stock price forecasting. The Symmetric Mean Absolute Percentage Error (SMAPE), Mean Absolute Scaled Error (MASE), and Overall Weighted Average (OWA) are used as metrics for M4 dataset [72, 42]. The formulations are shown in Eq. 5. Let $\mathbf{y}_t$ and $\widehat{\mathbf{y}}_t$ denote the ground-truth and the forecast at the $t$-th timestep, respectively. $H$ is the forecasting horizon. In M4 dataset, MASE is a standard scale-free metric, where $s$ is the periodicity of the data (*e.g.,* 12 for monthly recorded sub-dataset) [72]. MASE measures the scaled error *w.r.t.* the naïve predictor that simply copies the historical records of $s$ periods in the past. Instead, SMAPE scales the error by the average between the forecast and ground truth. Particularly, OWA is an M4-specific metric [81] that assigns different weights to each metric.

$$\text{MAE} = \frac{1}{H}\sum_{t=1}^{H}|\mathbf{y}_t - \widehat{\mathbf{y}}_t|, \qquad \text{MSE} = \frac{1}{H}\sum_{i=1}^{H}(\mathbf{y}_t - \widehat{\mathbf{y}}_t)^2,$$

$$\text{SMAPE} = \frac{200}{H}\sum_{i=1}^{H}\frac{|\mathbf{y}_t - \widehat{\mathbf{y}}_t|}{|\mathbf{y}_t| + |\widehat{\mathbf{y}}_t|}, \qquad \text{MASE} = \frac{1}{H}\sum_{i=1}^{H}\frac{|\mathbf{y}_t - \widehat{\mathbf{y}}_t|}{\frac{1}{H-s}\sum_{j=s+1}^{H}|\mathbf{y}_j - \mathbf{y}_{j-s}|}, \qquad (5)$$

$$\text{OWA} = \frac{1}{2}\left[\frac{\text{SMAPE}}{\text{SMAPE}_{\text{Naïve2}}} + \frac{\text{MASE}}{\text{MASE}_{\text{Naïve2}}}\right]$$

## C.3 Experiment Details

To verify the superiority of CCM in enhancing the performance of mainstream time series models, we select four popular and state-of-the-art models, including linear models such as TSMixer [7], DLinear [8], transformer-based model PatchTST [21] and convolution-based model TimesNet [13]. We build time series models using their official codes and optimal model configuration[1234].

In the data preprocessing stage, we apply reversible instance normalization [64] to ensure zero mean and unit standard deviation, avoiding the time series distribution shift. Forecasting loss is MSE for long-term forecasting datasets and the stock dataset. Instead, we use SMAPE loss for M4 dataset. We select Adam [82] with the default hyperparameter configuration for $(\beta_1, \beta_2)$ as (0.9, 0.999). An early-stopping strategy is used to mitigate overfitting. The experiments are conducted on a single NVIDIA RTX A6000 48GB GPU, with PyTorch [83] framework. We use the official codes and follow the best model configuration to implement the base models. Then we apply CCM to the base models, keeping the hyperparameters unchanged for model backbones. Experiment configurations, including the number of MLP layers in the cluster assigner, the layer number in the temporal module, hidden dimension, the best cluster number, and regularization parameter $\beta$ on nine real-world datasets are shown in Table 10.

Table 10: Experiment configuration.

| | # clusters | $\beta$ | # linear layers in MLP | hidden dimension | # layers (TSMixer) | # layers (PatchTST) | # layers (TimesNet) |
|---|---|---|---|---|---|---|---|
| ETTh1 | 2 | 0.3 | 1 | 128 | 2 | 2 | 3 |
| ETTm1 | 2 | 0.3 | 1 | 64 | 2 | 4 | 2 |
| ETTh2 | 2 | 0.3 | 1 | 64 | 2 | 4 | 3 |
| ETTm2 | 2 | 0.9 | 1 | 24 | 2 | 4 | 4 |
| Exchange | 2 | 0.9 | 1 | 32 | 2 | 4 | 3 |
| ILI | 2 | 0.9 | 1 | 36 | 2 | 6 | 3 |
| Weather | [2,5] | 0.5 | 2 | 64 | 4 | 3 | 3 |
| Electricity | [3,10] | 0.5 | 2 | 128 | 4 | 3 | 3 |
| Traffic | [3,10] | 0.5 | 2 | 128 | 4 | 3 | 3 |

## C.4 Comparison between CCM and Other Approach

Table 11: Full Results on Comparison between CCM and existing regularization method for enhanced performance on CI/CD strategies in terms of MSE metric. The best results are highlighted in **bold**.

| | | CD | CI | +PRReg | +CCM |
|---|---|---|---|---|---|
| ETTh1(48) | Linear | 0.402 | 0.345 | **0.342** | **0.342** |
| | Transformer | 0.861 | 0.655 | 0.539 | **0.518** |
| ETTh2(48) | Linear | 0.711 | 0.226 | 0.239 | **0.237** |
| | Transformer | 1.031 | 0.274 | **0.273** | 0.284 |
| ETTm1(48) | Linear | 0.404 | 0.354 | 0.311 | **0.310** |
| | Transformer | 0.458 | 0.379 | 0.349 | **0.300** |
| ETTm2(48) | Linear | 0.161 | 0.147 | **0.136** | 0.146 |
| | Transformer | 0.281 | 0.148 | 0.144 | **0.143** |
| Exchange(48) | Linear | 0.119 | 0.051 | **0.042** | **0.042** |
| | Transformer | 0.511 | 0.101 | **0.044** | 0.048 |
| Weather(48) | Linear | 0.142 | 0.169 | 0.131 | **0.130** |
| | Transformer | 0.251 | 0.168 | 0.180 | **0.164** |
| ILI(24) | Linear | 2.343 | 2.847 | 2.299 | **2.279** |
| | Transformer | 5.309 | 4.307 | 3.254 | **3.206** |
| Electricity(48) | Linear | **0.195** | 0.196 | 0.196 | **0.195** |
| | Transformer | 0.250 | 0.185 | 0.185 | **0.183** |

**Predict Residuals with Regularization**. Prior work [17] demonstrates that the main drawback of CD models is their inclination to generate sharp and non-robust forecasts, deviating from the actual trend. Thereby, Predict Residuals with Regularization (PRReg for simplicity), a specifically designed

---

[1]`https://github.com/yuqinie98/PatchTST`
[2]`https://github.com/cure-lab/LTSF-Linear`
[3]`https://github.com/google-research/google-research/tree/master/tsmixer`
[4]`https://github.com/thuml/TimesNet`

regularization objective, is proposed to improve the robustness of CD methods as follows.

$$\mathcal{L} = \frac{1}{N} \sum_{j=1}^{N} \mathcal{L}_F \left( f \left( X^{(j)} - A^{(j)} \right) + A^{(j)}, Y^{(j)} \right) + \lambda \Omega(f), \tag{6}$$

where the superscript $j$ indicates the sample index. $\mathcal{L}_F$ is MSE loss. $A^{(j)} = X_{L,:}^j$ represents the last values of each channel in $X^{(j)}$. Therefore, the objective encourages the model to generate predictions that are close to the nearby historical data and keep the forecasts smooth and robust. The regularization term $\Omega$ ($L_2$ norm in practice) further restricts the complexity of the model and ensures smoothness in the predictions. It was demonstrated that PRReg can achieve better performance than original CD and CI strategies [17]. We conduct extensive experiments on long-term forecasting benchmarks to compare PRReg and CCM. The full results are shown in Table 11. The numbers in parentheses next to the method represent the forecasting horizon. We set the length of the look-back window to 36 for ILI and 96 for other datasets for consistency with previous works [17]. The base models are linear model [8] and basic transformer [41]. The values in the PRReg column are the best results with any $\lambda$, collected from [17]. We observe from Table 11 that CCM successfully improves forecasting performance on original CI/CD strategies (or reached comparable results) in 13 out of 16 settings, compared with PRReg method.

### C.5   Results Analysis

We report the degree of multivariate correlation across multiple channels (measured by the average Pearson correlation coefficient) in Table 12 and Table 13. $r$ denotes the degree of multivariate correlation. Then the Pearson correlation coefficient between $r$ and the performance improvement rate is $0.258$ in long-term forecasting, indicating a weak correlation. It demonstrates that CCM tends to achieve a greater boost on datasets that are intrinsically correlated within channels. Compared with datasets used in long-term benchmarks, M4 demonstrates more significant correlations between time series. Therefore, capturing the correlation within the dataset in short-term cases potentially leads to greater improvement in the forecasting performance than in long-term cases.

Table 12: Multivariate intrinsic similarity for long-term forecasting datasets

| Dataset | ETTh1 | ETTm1 | ETTh2 | ETTm2 | Exchange | ILI | Weather | Electricity | Traffic |
|---|---|---|---|---|---|---|---|---|---|
| Correlation $r$ | 0.1876 | 0.1717 | 0.3224 | 0.328 | 0.3198 | 0.508 | 0.1169 | 0.5311 | 0.6325 |

Table 13: Intrinsic similarity for short-term forecasting datasets

| Dataset | M4 Monthly | M4 Daily | M4 Yearly | M4 Hourly | M4 Quarterly | M4 Weekly |
|---|---|---|---|---|---|---|
| Correlation $r$ | 0.62 | 0.646 | 0.712 | 0.55 | 0.671 | 0.653 |

### C.6   Error Bar

Experimental results in this paper are averaged from five runs with different random seeds. We report the standard deviation for base models and CCM-enhanced versions on long-term forecasting datasets in Table 14, M4 dataset in Table 15 and stock dataset in Table 16.

## D   Ablation Study

### D.1   Influence of Cluster Ratio

The number of clusters is an important hyperparameter in the CCM method. To verify the effectiveness of our design, we conduct an ablation study for all base models on four long-term forecasting datasets. The full results are shown in Table 17. We tune different cluster ratios, defined as the ratio of the number of clusters to the number of channels. *Original* means the original base model without any channel clustering mechanism. *0.0* indicates grouping all channels into the same cluster. We make the following observations. (1) For most cases, the channel clustering module (CCM) with any number of clusters greater than 1 consistently improves the forecasting performance upon base models. (2) For

Table 14: Standard deviation of Table 2 on long-term forecasting benchmarks. The forecasting horizon is 96.

| | | ETTh1 | ETTm1 | ETTh2 | ETTm2 | Exchange | ILI | Weather | Electricity | Traffic |
|---|---|---|---|---|---|---|---|---|---|---|
| TSMixer | MSE | 0.361±0.001 | 0.285±0.001 | 0.284±0.001 | 0.171±0.001 | 0.089±0.004 | 1.914±0.031 | 0.149±0.008 | 0.142±0.002 | 0.376±0.006 |
| | MAE | 0.392±0.001 | 0.339±0.001 | 0.343±0.002 | 0.260±0.001 | 0.209±0.009 | 0.879±0.009 | 0.198±0.009 | 0.237±0.004 | 0.264±0.005 |
| +CCM | MSE | 0.365±0.001 | 0.283±0.002 | 0.278±0.001 | 0.167±0.001 | 0.085±0.006 | 1.938±0.015 | 0.147±0.007 | 0.139±0.005 | 0.375±0.006 |
| | MAE | 0.393±0.001 | 0.337±0.002 | 0.338±0.002 | 0.260±0.001 | 0.206±0.011 | 0.874±0.012 | 0.194±0.009 | 0.235±0.004 | 0.262±0.005 |
| DLinear | MSE | 0.375±0.002 | 0.299±0.001 | 0.289±0.001 | 0.167±0.001 | 0.088±0.006 | 2.215±0.031 | 0.192±0.011 | 0.153±0.004 | 0.411±0.006 |
| | MAE | 0.399±0.001 | 0.343±0.001 | 0.353±0.001 | 0.260±0.001 | 0.215±0.010 | 1.081±0.009 | 0.250±0.008 | 0.239±0.005 | 0.284±0.005 |
| +CCM | MSE | 0.371±0.001 | 0.298±0.001 | 0.285±0.001 | 0.166±0.002 | 0.085±0.006 | 1.935±0.034 | 0.187±0.015 | 0.142±0.003 | 0.411±0.005 |
| | MAE | 0.393±0.001 | 0.343±0.002 | 0.348±0.02 | 0.258±0.002 | 0.214±0.013 | 0.935±0.012 | 0.245±0.020 | 0.247±0.006 | 0.282±0.004 |
| PatchTST | MSE | 0.375±0.003 | 0.294±0.003 | 0.278±0.003 | 0.174±0.003 | 0.094±0.008 | 1.593±0.016 | 0.149±0.008 | 0.138±0.004 | 0.360±0.005 |
| | MAE | 0.398±0.004 | 0.351±0.004 | 0.340±0.004 | 0.261±0.003 | 0.216±0.012 | 0.757±0.015 | 0.198±0.012 | 0.233±0.005 | 0.249±0.005 |
| +CCM | MSE | 0.371±0.002 | 0.289±0.005 | 0.274±0.006 | 0.168±0.003 | 0.088±0.006 | 1.561±0.021 | 0.147±0.008 | 0.136±0.002 | 0.357±0.007 |
| | MAE | 0.396±0.003 | 0.338±0.005 | 0.336±0.006 | 0.256±0.003 | 0.208±0.009 | 0.750±0.009 | 0.197±0.013 | 0.231±0.006 | 0.246±0.006 |
| TimesNet | MSE | 0.384±0.005 | 0.338±0.006 | 0.340±0.005 | 0.187±0.005 | 0.107±0.009 | 2.317±0.024 | 0.172±0.011 | 0.168±0.002 | 0.593±0.010 |
| | MAE | 0.402±0.005 | 0.375±0.006 | 0.374±0.005 | 0.267±0.003 | 0.234±0.013 | 0.934±0.010 | 0.220±0.013 | 0.272±0.005 | 0.321±0.008 |
| +CCM | MSE | 0.380±0.004 | 0.335±0.005 | 0.336±0.003 | 0.189±0.003 | 0.105±0.006 | 2.139±0.038 | 0.169±0.015 | 0.158±0.003 | 0.554±0.009 |
| | MAE | 0.400±0.004 | 0.371±0.006 | 0.371±0.005 | 0.270±0.005 | 0.231±0.010 | 0.936±0.018 | 0.215±0.024 | 0.259±0.006 | 0.316±0.008 |

Table 15: Standard deviation of Table 6 on M4 dataset

| Model | | TSMixer | + CCM | DLinear | + CCM | PatchTST | + CCM | TimesNet | + CCM |
|---|---|---|---|---|---|---|---|---|---|
| Yearly | SMAPE | 0.122 | 0.130 | 0.087 | 0.089 | 0.135 | 0.134 | 0.168 | 0.162 |
| | MASE | 0.022 | 0.022 | 0.019 | 0.017 | 0.018 | 0.021 | 0.0017 | 0.017 |
| | OWA | 0.002 | 0.002 | 0.002 | 0.002 | 0.007 | 0.009 | 0.010 | 0.011 |
| Quaterly | SMAPE | 0.101 | 0.103 | 0.100 | 0.100 | 0.079 | 0.073 | 0.106 | 0.105 |
| | MASE | 0.016 | 0.016 | 0.015 | 0.015 | 0.008 | 0.009 | 0.013 | 0.011 |
| | OWA | 0.008 | 0.007 | 0.006 | 0.008 | 0.013 | 0.016 | 0.009 | 0.009 |
| Monthly | SMAPE | 0.113 | 0.113 | 0.110 | 0.111 | 0.122 | 0.120 | 0.120 | 0.134 |
| | MASE | 0.013 | 0.015 | 0.009 | 0.013 | 0.017 | 0.019 | 0.011 | 0.012 |
| | OWA | 0.001 | 0.001 | 0.002 | 0.001 | 0.003 | 0.002 | 0.004 | 0.004 |
| Others | SMAPE | 0.113 | 0.110 | 0.126 | 0.128 | 0.137 | 0.130 | 0.129 | 0.125 |
| | MASE | 0.024 | 0.026 | 0.036 | 0.031 | 0.023 | 0.025 | 0.028 | 0.025 |
| | OWA | 0.011 | 0.013 | 0.009 | 0.008 | 0.023 | 0.019 | 0.024 | 0.020 |
| Avg. | SMAPE | 0.113 | 0.115 | 0.111 | 0.103 | 0.136 | 0.134 | 0.148 | 0.153 |
| | MASE | 0.027 | 0.025 | 0.021 | 0.017 | 0.026 | 0.021 | 0.027 | 0.042 |
| | OWA | 0.006 | 0.004 | 0.004 | 0.004 | 0.013 | 0.016 | 0.021 | 0.036 |

Channel-Independent base models, such as DLinear and PatchTST, grouping all channels into one cluster results in a channel fusion at the last layer, leading to a degradation in forecasting performance compared to the original CI models. (3) For most cases, the cluster ratio in the range of $[0.2, 0.6]$ typically results in the best performance.

## D.2 Influence of Look-back Window Length

In this section, we conduct additional ablation experiments to investigate the effect of different look-back window lengths in the newly collected stock dataset, which determines how much historical information the time series model uses to make short-term forecasts. Specifically, the ablation study helps identify the risk of overfitting or underfitting based on the chosen look-back window length. An overly long window may lead to overfitting, while a short window may cause underfitting. Table 18 display the forecasting performance on forecasting horizon 7 and 24. The length of the look-back window ranges from two to four times the forecasting horizon. From Table 18, we make the following observations. (1) CCM effectively improves the base models' forecasting performance in 24 cases across different base models, forecasting horizons, and look-back window lengths consistently. (2) Especially, CCM achieves better enhancement when the look-back window is shorter.

## D.3 Influence of Different Clustering Steps

We conducted an ablation study on different clustering steps to investigate its effect on downstream performance, reported in Table 19. The ablation study follows the setting in Table 5. Random means we randomly assign each channel to a cluster, leading to worse clustering quality. K-Means means using the k-means algorithm to replace our clustering step, leading to suboptimal prototype embedding. The proposed CCM is essentially an advanced variant of K-Means with learnable prototype embedding and cross-attention mechanism. Results show that both clustering quality and

Table 16: Standard deviation of Table 6 on stock dataset

| Horizon | Metric | TSMixer | + CCM | DLinear | + CCM | PatchTST | + CCM | TimesNet | + CCM |
|---|---|---|---|---|---|---|---|---|---|
| 7 | MSE | 0.001 | 0.001 | 0.001 | 0.001 | 0.003 | 0.003 | 0.004 | 0.004 |
|  | MAE | 0.001 | 0.001 | 0.001 | 0.001 | 0.003 | 0.002 | 0.003 | 0.003 |
| 24 | MSE | 0.002 | 0.002 | 0.001 | 0.002 | 0.005 | 0.004 | 0.005 | 0.005 |
|  | MAE | 0.001 | 0.001 | 0.001 | 0.001 | 0.003 | 0.002 | 0.003 | 0.004 |

Table 17: Sensitivity of cluster ratio in terms of MSE metric. The forecasting horizon is 96. *0.0* means grouping all channels into the same cluster. *original* means the base model without the CCM mechanism.

| | Cluster Ratio | Original | 0.0 | 0.2 | 0.4 | 0.6 | 0.8 | 1.0 |
|---|---|---|---|---|---|---|---|---|
| ETTh1 | TSMixer | 0.361 | 0.361 | 0.362 | 0.365 | 0.363 | 0.364 | 0.366 |
|  | DLinear | 0.375 | 0.378 | 0.371 | 0.371 | 0.372 | 0.372 | 0.371 |
|  | PatchTST | 0.375 | 0.380 | 0.372 | 0.371 | 0.373 | 0.376 | 0.375 |
|  | TimesNet | 0.384 | 0.384 | 0.380 | 0.383 | 0.385 | 0.385 | 0.388 |
| ETTm1 | TSMixer | 0.285 | 0.285 | 0.283 | 0.283 | 0.284 | 0.285 | 0.286 |
|  | DLinear | 0.299 | 0.303 | 0.298 | 0.298 | 0.299 | 0.300 | 0.300 |
|  | PatchTST | 0.294 | 0.298 | 0.292 | 0.289 | 0.289 | 0.293 | 0.293 |
|  | TimesNet | 0.338 | 0.338 | 0.337 | 0.335 | 0.336 | 0.335 | 0.337 |
| Exchange | TSMixer | 0.089 | 0.089 | 0.086 | 0.087 | 0.088 | 0.090 | 0.092 |
|  | DLinear | 0.088 | 0.093 | 0.088 | 0.087 | 0.085 | 0.089 | 0.089 |
|  | PatchTST | 0.094 | 0.095 | 0.089 | 0.088 | 0.088 | 0.091 | 0.093 |
|  | TimesNet | 0.107 | 0.107 | 0.105 | 0.105 | 0.107 | 0.107 | 0.107 |
| Electricity | TSMixer | 0.142 | 0.142 | 0.139 | 0.139 | 0.140 | 0.143 | 0.143 |
|  | DLinear | 0.153 | 0.160 | 0.143 | 0.142 | 0.143 | 0.147 | 0.150 |
|  | PatchTST | 0.138 | 0.142 | 0.136 | 0.136 | 0.138 | 0.140 | 0.140 |
|  | TimesNet | 0.168 | 0.168 | 0.160 | 0.159 | 0.167 | 0.168 | 0.169 |

prototype embedding will affect the downstream performance. Instead, CCM generates high-quality channel clustering results, compared with random assignment and K-Means clustering.

# E Visualization Results

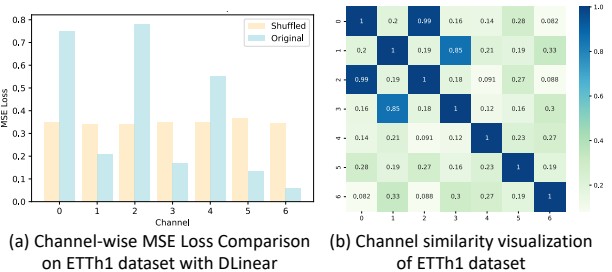

(a) Channel-wise MSE Loss Comparison on ETTh1 dataset with DLinear

(b) Channel similarity visualization of ETTh1 dataset

Figure 6: (a) Channel-wise forecasting performance and (b) Channel similarity on ETTh1 dataset illustrate the correlation between model performance and intrinsic similarity

## E.1 Channel-wise Performance and Channel Similarity

Figure 6 illustrates the channel-wise forecasting performance in terms of MSE metric and channel similarity on ETTh1 dataset with DLinear. We use the model's performance difference on the original dataset and the randomly shuffled dataset as the model's fitting ability on a specific channel. Note that MSE loss is computed on channels that have been standardized, which means that any scaling differences between them have been accounted for. Figure 6 highlights a noteworthy pattern: when two channels exhibit a higher degree of similarity, there tends to be a corresponding similarity in the performance on these channels. This observation suggests that channels with similar characteristics tend to benefit similarly from the optimization. It implies a certain level of consistency in the improvement of MSE loss when dealing with similar channels.

Table 18: Short-term forecasting on stock dataset with different look-back window length in {14, 21, 28}. The forecasting length is 7. The best results with the same base model are underlined. **Bold** means CCM successfully enhances forecasting performance over the base model.

| Forecast | 7 | | | | | | 24 | | | | | |
|---|---|---|---|---|---|---|---|---|---|---|---|---|
| Input Length | 14 | | 21 | | 28 | | 48 | | 72 | | 96 | |
| | MSE | MAE | MSE | MAE | MSE | MAE | MSE | MAE | MSE | MAE | MSE | MAE |
| TSMixer | 0.947 | 0.806 | 0.974 | 0.816 | 0.939 | 0.807 | 1.007 | 0.829 | 1.016 | 0.834 | 1.100 | 0.856 |
| + CCM | **0.896** | **0.778** | **0.954** | **0.808** | **0.938** | **0.806** | **0.991** | **0.817** | **1.006** | **0.824** | **1.078** | **0.851** |
| DLinear | 1.003 | 0.834 | 0.995 | 0.833 | 0.992 | 0.831 | 0.998 | 0.832 | 0.996 | 0.832 | 0.998 | 0.832 |
| + CCM | **0.897** | **0.778** | **0.904** | **0.782** | **0.883** | **0.774** | **0.921** | **0.786** | **0.917** | **0.781** | **0.969** | **0.798** |
| PatchTST | 0.933 | 0.804 | 0.896 | **0.771** | 0.926 | 0.794 | 0.976 | 0.793 | 0.951 | 0.790 | 0.930 | 0.789 |
| + CCM | **0.931** | **0.758** | **0.892** | 0.771 | **0.924** | **0.790** | **0.873** | **0.767** | **0.860** | **0.759** | **0.880** | **0.765** |
| TimesNet | 0.943 | 0.816 | 0.934 | 0.803 | 0.930 | 0.802 | 0.998 | 0.830 | 1.003 | 0.818 | 1.013 | 0.821 |
| + CCM | **0.926** | **0.796** | **0.911** | **0.789** | **0.915** | **0.793** | **0.937** | **0.789** | **0.974** | **0.803** | **0.979** | **0.804** |
| Imp. (%) | 4.492 | 4.590 | 3.527 | 2.211 | 3.230 | 2.152 | 6.492 | 3.798 | 5.344 | 3.271 | 3.409 | 2.445 |

Table 19: Ablation on different clustering steps on ETTh1 and ETTm1 based on Linear and Transformer architecture.

| | | CD | CI | Random | K-Means | CCM |
|---|---|---|---|---|---|---|
| ETTh1 | Linear | 0.402 | 0.345 | 0.389 | 0.357 | **0.342** |
| | Transformer | 0.861 | 0.655 | 0.746 | 0.542 | **0.518** |
| ETTm1 | Linear | 0.404 | 0.354 | 0.371 | 0.326 | **0.310** |
| | Transformer | 0.458 | 0.379 | 0.428 | 0.311 | **0.300** |

# F  Discussion

This paper presents the Channel Clustering Module (CCM) for enhanced performance of time series forecasting models, aiming to balance the treatment of individual channels while capturing essential cross-channel interactions. Despite its promising contributions, there still exist limitations and directions for future work that warrant consideration.

**Limitation**. While CCM shows improvements in forecasting, its scalability to extremely large datasets remains to be tested. Moreover, the clustering and embedding processes in CCM introduce additional computational overhead. The efficiency of CCM in real-time forecasting scenarios, where computational resources are limited, requires further optimization.

**Future Works**. Future research can focus on adapting CCM to specific domains, such as biomedical signal processing or geospatial data analysis, where external domain-specific knowledge can be involved in the similarity computation. Furthermore, exploring alternative approaches to develop a dynamical clustering mechanism within CCM could potentially enhance forecasting efficacy. It is also worth examining the effectiveness of CCM in contexts beyond time series forecasting.

**Social Impact**. The Channel Clustering Module (CCM) presented in this paper holds significant potential for positive social impact. By improving the accuracy and efficiency of forecasting, CCM can benefit a wide range of applications critical to society. For instance, in healthcare, CCM could enhance the prediction of patient outcomes, leading to better treatment planning and resource allocation. Additionally, in financial markets, CCM could aid in predicting market trends, supporting informed decision-making and potentially reducing economic risks for individuals and organizations. Overall, the development and refinement of CCM could potentially enhance the quality of life and promote societal well-being.

