# OpenReview forum: "From Similarity to Superiority: Channel Clustering for Time Series Forecasting"
_NeurIPS.cc/2024/Conference — NeurIPS 2024 poster_

### Official Review · Reviewer_SRyH · 2024-06-29

**Soundness:** 2
**Presentation:** 3
**Contribution:** 2
**Rating:** 5
**Confidence:** 4

**Summary:**

The paper introduces the Channel Clustering Module (CCM), a novel approach to enhance time series forecasting models. CCM addresses the limitations of traditional Channel-Independent (CI) and Channel-Dependent (CD) strategies by dynamically clustering channels based on their intrinsic similarities. This approach allows the model to balance individual channel treatment with capturing essential cross-channel dependencies, leading to improved forecasting performance. CCM is adaptable to various time series models and demonstrates its effectiveness through extensive experiments on multiple real-world datasets.

**Strengths:**

- The originality of CCM lies in its novel approach to channel clustering for time series forecasting, addressing the limitations of existing strategies.
- The quality of the work is evident in the well-designed experiments and the clear presentation of results, showcasing the effectiveness of CCM across different datasets and models.
- The paper’s clarity in explaining the concept, methodology, and results enhances its readability and understanding.

**Weaknesses:**

- The paper could benefit from more detailed discussions on the selection of similarity metrics and the impact of hyperparameters on performance.
- The computational efficiency of CCM, especially in large-scale applications, is not extensively discussed.

**Questions:**

- Where is the official code?
- In terms of thinking, the principal component analysis method is actually similar to the proposed method. Please provide a detailed explanation.

**Limitations:**

The limitation of the Channel Clustering Module (CCM) outlined in the paper includes its scalability to extremely large datasets and the computational overhead introduced by the clustering and embedding processes. While CCM shows improvements in forecasting, its efficiency in real-time forecasting scenarios with limited computational resources remains to be optimized. Additionally, the clustering and embedding processes in CCM introduce additional computational overhead, which could be a concern in scenarios where computational efficiency is critical.

---

> ### Author Rebuttal · Authors · 2024-08-07
>
> Thanks for your valuable feedback and positive comments.  We address the potential concerns as follows.
>
> >The computational efficiency of CCM, especially in large-scale applications, is not extensively discussed.
>
> As discussed in Sec.4.3, the computational complexity of CCM scales linearly with the number of channels $C$, which is computationally efficient for real-world deployment. Our extensive experiments, including large-scale datasets such as traffic (15,122,928 observations) and stock (13,900,000 observations), provide strong evidence for CCM's promising scalability on extremely large-scale datasets: 1) CCM consistently **reduces model complexity for CI models** (e.g. DLinear and PatchTST), regardless of dataset scale or size (Fig. 5); 2) The **linear scaling w.r.t channel count** enables efficient handling of high-dimensional data, crucial for many practical applications; and 3) our experiments reveal that **the performance gains achieved by CCM are maintained across different dataset sizes**, suggesting that the method's benefits are robust and not limited to specific data scales. This demonstrates the efficiency and scalability of CCM, making it promising for extremely large-scale real-world deployments.
>
> Furthermore, we've identified several strategies to optimize the scalability of CCM, including leveraging parallel and distributed frameworks for cluster assigner training and applying sampling on time series similarity computation to optimize computational resources. Algorithmic optimizations, such as efficient and fast attention with linear time complexity further support CCM's scalability. However, we would like to emphasize that these techniques are compatible and orthogonal to the contribution of this manuscript and require substantial additional research that would expand beyond our current scope. We appreciate your suggestion and will expand our discussion on computational efficiency in the revised version.
>
> >Where is the official code?
>
> As we mentioned in Line677 (Appendix C.3), the code is available at the following anonymous link: https://anonymous.4open.science/r/TimeSeriesCCM-4E83. We will have an official GitHub repository once the paper is accepted.
>
> >The principal component analysis method is actually similar to the proposed method. Please provide a detailed explanation.
>
> PCA and the proposed CCM serve fundamentally **different purposes and operate on distinct principles**.
>
> PCA is primarily a dimensionality reduction technique that transforms high-dimensional data into a lower-dimensional space, preserving maximum variance. It does not inherently cluster data but rather restructures it into orthogonal components. In contrast, CCM is specifically designed for time series forecasting, focusing on grouping channels based on their intrinsic similarities to enhance prediction accuracy and interpretability. CCM dynamically clusters channels into cohesive groups, leveraging these similarities to capture complex inter-channel dependencies. It enhances forecasting by allowing models to treat clusters as coherent entities, thus improving both individual channel fit and cross-channel interactions.
>
> Essentially, while PCA emphasizes variance capture and dimensionality reduction, CCM prioritizes similarity-based clustering to optimize time series analysis and forecasting.
>
> >Detailed discussions on the selection of similarity metrics and the impact of hyperparameters on performance would benefit the paper.
>
> The selection of similarity metrics and other alternatives are discussed thoroughly in Appendix A.1. We also conducted ablation studies on cluster ratio and look-back window length to investigate their impact on model performance. Please refer to Appendix D for detailed results and analysis.
>
> We hope that the above clarification improves your confidence in our work. Let us know if you have any further questions/concerns.

---

> > ### Comment · Reviewer_SRyH · 2024-08-13
> >
> > Thank you for the rebuttal. The clarifications provided have mitigated some concerns.

---

> ### Author Response · Authors · 2024-08-13
> **We would like to hear from Reviewer SRyH**
>
> Dear Reviewer SRyH,
>
> As the discussion period is close to end, we would like to follow up and ensure that our responses have adequately addressed your concerns.
>
> We sincerely appreciate your comments and suggestions, which have significantly contributed to improving our paper. In response to your valuable feedback, we have added more discussion on the model efficiency and clarified the difference between PCA and our proposed CCM. We would be more than happy to further discuss if there are any remaining questions. Thanks again for your time and consideration.
>
> Regards,\
> Authors

---

> > ### Comment · Reviewer_SRyH · 2024-08-13
> >
> > I will complete my response promptly. Thank you for the reminder.

---

### Official Review · Reviewer_rxdY · 2024-07-04

**Soundness:** 3
**Presentation:** 3
**Contribution:** 3
**Rating:** 8
**Confidence:** 4

**Summary:**

Time series forecasting has been a topic of interest, with previous studies exploring different strategies. The Channel-Independent (CI) strategy treats channels individually, improving forecasting performance but lacking generalization and ignoring channel interactions. On the other hand, the Channel-Dependent (CD) strategy combines channels indiscriminately, leading to oversmoothing and reduced accuracy. A channel strategy is needed that balances individual treatment and essential interactions. Based on the correlation between performance and channel mixing, a novel Channel Clustering Module (CCM) was developed. CCM groups channels with intrinsic similarities and utilizes cluster information, combining the advantages of CD and CI. Experimental results show that CCM enhances the performance of CI and CD models, enables zero-shot forecasting, and improves interpretability of complex models by uncovering intrinsic patterns among channels.

**Strengths:**

1. The proposed model-agnostic method CCM achieves optimal performance between single-channel and cross-channel modeling, and it can be integrated into existing time series prediction models to enhance their performance.
2. By learning prototypes from clusters, CCM facilitates zero-shot forecasting on unseen samples, whether in univariate or multivariate scenarios.
3. The author integrated CCM into four mainstream time series prediction models on multiple different datasets. The experimental results demonstrate that in most cases, CCM can bring about significant performance improvements.

**Weaknesses:**

1. The experimental section involves a limited number of baseline methods, for example, SOTA LLM-based time series prediction models [1, 2] were not selected.
2. I noticed that CCM introduces additional model complexity, as it has an independent Feed Forward layer for each cluster. When the value of K is large, this may result in an excessive number of Feed Forward layers, leading to a significant increase in space complexity. On the other hand, the time complexity of CCM is linearly related to K and C. For certain datasets with a higher number of channels (e.g., Traffic), CCM may noticeably increase the time complexity of the base model. Some of the results in Figure 5 of the Appendix hint at this issue.
3. According to Table 14, the improvements brought by CCM to the base model are sometimes overshadowed by the perturbation caused by randomness. This may indicate that CCM has certain limitations.

[1] Zhou, Tian, et al. "One fits all: Power general time series analysis by pretrained lm." Advances in neural information processing systems 36 (2023): 43322-43355.
[2] Jin, Ming, et al. "Time-llm: Time series forecasting by reprogramming large language models." arXiv preprint arXiv:2310.01728 (2023).

**Questions:**

1. Can the author provide the percentage increase in runtime and memory usage after integrating CCM for each method on the Traffic dataset?
2. Can a certain hyperparameter analysis be conducted for the parameter β (i.e., β ∈ {0, 0.25, 0.5, 0.75, 1.0}) to demonstrate the impact of the Cluster Loss on model performance?
3.  Can the author provide a significance test (i.e., p-value test) for the experimental results, especially for cases where the values in Table 4 are close? As in some cases, the improvements are minimal, and the observed performance gains might be due to randomness rather than the effectiveness of CCM.

**Limitations:**

The author elaborates on the limitations of their work. The main limitations are as follows: CCM does not outperform CI/CD in a few cases in the long-term forecasting benchmark; CCM’s scalability to extremely large datasets remains to be tested.

---

> ### Author Rebuttal · Authors · 2024-08-07
>
> Thanks for your valuable feedback and positive comments.  We address the potential concerns as follows.
>
> >Some LLM-based models were not selected as baselines.
>
> After careful consideration, we've decided not to include LLM-based time series models in this manuscript for several reasons. Firstly, many LLM-based models treat multivariate time series as independent univariate series, which doesn't align with our core assumption about the importance of cross-channel interactions [1,2]. Secondly, some LLM-based methods that concatenate variates into a single long series [3] are incompatible with CCM's clustering approach, as the clustering will destroy the attention calculation along the single univariate time series. While we acknowledge the growing importance of LLM-based methods in time series forecasting, investigating clustering effects on these models requires substantial additional research that would expand beyond our current scope. We believe this warrants a separate, dedicated study. Our focus on traditional deep learning models allows for a thorough analysis of CCM's effectiveness within an established framework. We are more than happy to add discussions in our revised version.
>
> [1] One fits all: Power general time series analysis by pretrained lm\
> [2] Time-llm: Time series forecasting by reprogramming large language models\
> [3] Unified Training of Universal Time Series Forecasting Transformers
>
> >CCM introduces additional space and time complexity. CCM’s scalability remains to be tested.
>
> As demonstrated in Sec 4.3, CCM's computational complexity scales linearly with the number of channels $C$, ensuring efficient real-world applications. Our extensive experiments provide strong evidence for CCM's promising scalability on extremely large-scale datasets:
> - CCM consistently reduces model complexity for CI models (e.g. DLinear and PatchTST), regardless of dataset scale or size (Fig. 5).
> - The linear scaling w.r.t channel count enables efficient handling of high-dimensional data, crucial for many practical applications.
> - CCM's performance gains persist across various dataset sizes, indicating robust benefits independent of data scale.
>
> These findings collectively underscore CCM's efficiency and scalability, making it a highly promising approach for large-scale real-world deployments.
>
> >The improvements brought by CCM may sometimes be caused by randomness. Can the author provide a significance test?
>
> We conducted a significance test using p-values for the ETTh1 dataset in Table 4, which represents the scenario where CCM shows a minimal performance boost among our experiments. The p-value test results are as follows (values lower than 0.05 are in bold):
> |ETTh1|96|192|336|720|
> |:-:|:-:|:-:|:-:|:-:|
> |TSMixer |-|  **0.0046**  |**0.007**|**0.011** |
> |  DLinear |**0.034** |0.093|**0.041** | **0.009** |
> | PatchTST |-|**1.12E-05** |**0.0004** |**0.002** |
> | TimesNet |0.089|**0.0019**|**0.043**|0.064|
>
> CCM's improvement effect is significant in general. Even in this "worst-case" scenario for CCM (i.e., ETTh1 dataset), we observe significant improvements across multiple forecast horizons and base models. Specifically, the majority of the p-values (11 out of 14) are below the conventional significance threshold of 0.05, indicating statistical significance. Despite a few cases where the p-values are above 0.05, the overall trend strongly supports the effectiveness of CCM. We will include a more comprehensive significance test in our revised version.
>
> >The percentage increase in runtime and memory usage on the Traffic dataset?
>
> We provide the percentage increase in model size and runtime on the Traffic dataset based on TimesNet and TSMixer (two CD models) as follows.
>
> $\Delta$ Param (%):
> | BaseModel / #clusters |2|3|4|5|
> |-|:-:|:-:|:-:|:-:|
> |TimesNet| 0.052 |  0.088 |0.125 | 0.162 |
> |TSMixer| 9.845 |18.420|26.995 |35.570|
>
> $\Delta$ iter. time (%):
> | BaseModel / #clusters |2|3|4|5|
> |-|:-:|:-:|:-:|:-:|
> |TimesNet|2.907 |3.632|5.291|5.777|
> |TSMixer| 25.746 | 41.350| 53.352 | 64.462|
>
> We observe that CCM represents a minimal increase in memory usage and runtime on TimesNet, demonstrating its parameter efficiency when applied to TimesNet. For TSMixer, there is a greater increase compared to TimesNet. However, it's important to note that TSMixer is inherently a more compact model. The absolute increase in parameters remains relatively small and reasonable. It's worth noting that these increases should be considered in the context of the performance improvements that CCM brings, which justifies this modest increase in model size. Moreover,  when applied to CI models, (eg, DLinear and PatchTST), CCM consistently reduces the model complexity and runtime, further highlighting its versatility and efficiency.
>
> >A hyperparameter analysis for the parameter $\beta$?
>
> We conducted a hyperparameter analysis for $\beta$, ranging from 0.1 to 2.0 on ETTh2 dataset (H=720) as follows.
> |$\beta$ |w/o CCM|0.1|0.3|0.5|1.0|2.0|
> |-|:-:|:-:|:-:|:-:|:-:|:-:|
> |TSMixer | 0.445±0.006 | 0.442±0.006 |**0.438±0.003**|0.442±0.009|0.439±0.005 |0.440±0.007|
> |DLinear | 0.601±0.008 | 0.537±0.013 |**0.499±0.012**|0.501±0.010|0.510±0.008|0.524±0.013|
> |PatchTST | 0.381±0.005 | 0.381±0.004 | 0.379±0.004 |**0.378±0.007**| 0.379±0.004 |0.381±0.009|
> |TimesNet | 0.462±0.009 | 0.460±0.006 | **0.457±0.003** |**0.457±0.003**| 0.460±0.004 |0.468±0.008|
>
> We observe that the introduction of CCM (with appropriate $\beta$) consistently improves performance compared to the baseline without CCM. The optimal $\beta$ value varies slightly across different architectures, but generally falls in the range of 0.3 to 0.5. Moreover, we observe that TSMixer, PatchTST, and TimesNet show more stable performance than DLinear across different $\beta$ values. We will include the sensitivity analysis on $\beta$ in our revised version.
>
> We hope that the clarification improves your confidence in our work. Let us know if you have any further concerns.

---

> ### Author Response · Authors · 2024-08-12
> **We would like to hear from Reviewer rxdY**
>
> Dear Reviewer rxdY,
>
> As the discussion period is close to end, we would like to follow up and ensure that our responses have adequately addressed your concerns. We sincerely appreciate your comments and suggestions, which have significantly contributed to improving our paper. In response to your valuable feedback, we conducted additional experiments on significance test, complexity evaluation and hyperparameter ablation study. We would be more than happy to further discuss if there are any remaining questions. Thanks again for your time and consideration.
>
> Regards,\
> Authors

---

> > ### Comment · Reviewer_rxdY · 2024-08-13
> >
> > Thanks for the authors' rebuttal. Since the authors well addressed my concerns, I will raise my score.

---

### Official Review · Reviewer_yXcm · 2024-07-11

**Soundness:** 3
**Presentation:** 3
**Contribution:** 3
**Rating:** 4
**Confidence:** 3

**Summary:**

The paper presents a new Channel Clustering Module (CCM) for time series forecasting, which dynamically groups channels based on intrinsic similarities to balance the strengths of Channel-Independent (CI) and Channel-Dependent (CD) strategies. CCM improves forecasting accuracy by enhancing model generalization and capturing essential cross-channel interactions, achieving significant performance gains in both long-term and short-term forecasting. The module also supports zero-shot forecasting and improves the interpretability of complex time series models. Extensive experiments demonstrate CCM's effectiveness and adaptability across various mainstream time series models.

**Strengths:**

1. The paper introduces a new Channel Clustering Module (CCM) that balances individual channel treatment and cross-channel dependencies, combining the strengths of Channel-Independent (CI) and Channel-Dependent (CD) strategies.

2. CCM enables zero-shot forecasting, leveraging learned prototypes to handle unseen samples effectively.

**Weaknesses:**

1. The introduction of CCM increases the model's complexity, particularly for original CD models, which may result in higher computational overhead.

2. The paper acknowledges that the scalability of CCM to extremely large datasets remains untested, which could be a limitation for practical applications requiring the processing of large-scale data.

**Questions:**

1. How can the similarity metric be further refined to enhance the clustering quality and address performance variations across different domains?

2. What strategies can be employed to test and ensure the scalability of CCM for extremely large datasets?

3. How does CCM perform in real-world applications with diverse and dynamic datasets, and what adjustments are necessary to optimize its performance in such scenarios?

4. What techniques can be implemented to mitigate the increased computational overhead introduced by CCM, especially for models with high channel and cluster counts?

**Limitations:**

1. CCM does not outperform CI/CD strategies in a few cases, possibly due to the underlying channel relationships in certain real-world domains not aligning well with the similarity metric used by CCM.

2. The increased model complexity introduced by CCM may lead to higher computational costs, particularly for models with numerous channels and clusters.

---

> ### Author Rebuttal · Authors · 2024-08-07
>
> Thanks for your valuable feedback, which significantly improves the quality of this work. We also address below the potential concerns.
>
> >The introduction of CCM increases the model's complexity. The scalability of CCM to extremely large datasets remains untested. What strategies to ensure the scalability of CCM? What techniques to mitigate the increased computational overhead of CCM?
>
> As discussed in Sec.4.3, the computational complexity of CCM scales linearly with the number of channels $C$, which is computationally efficient for real-world deployment. Our extensive experiments, including large-scale datasets such as traffic (15,122,928 observations) and stock (13,900,000 observations), provide strong evidence for CCM's promising scalability on extremely large-scale datasets:
> - CCM consistently **reduces model complexity for CI models** (e.g. DLinear and PatchTST), regardless of dataset scale or size (Fig. 5).
> - The **linear scaling w.r.t channel count** enables efficient handling of high-dimensional data, crucial for many practical applications.
> - Our experiments reveal that the **performance gains achieved by CCM are maintained across different dataset sizes**, suggesting that the method's benefits are robust and not limited to specific data scales.
>
> This demonstrates the efficiency and scalability of CCM, making it promising for extremely large-scale real-world deployments.
>
> Furthermore, we've identified several strategies to optimize the scalability of CCM, including leveraging parallel and distributed frameworks for cluster assigner training and applying sampling on time series similarity computation to optimize computational resources. Algorithmic optimizations, such as efficient and fast attention with linear time complexity further support CCM's scalability. However, we would like to emphasize that **these techniques are compatible and orthogonal to the contribution of this manuscript** and deserve substantial additional research that would expand beyond our current scope. We appreciate your suggestion and will expand our discussion on computational efficiency in the revised version.
>
> >How can the similarity metric be further refined to enhance the clustering quality and address performance variations across different domains?
>
> The similarity metrics and alternatives are discussed thoroughly in Appendix A.1. We select the Euclidean-based similarity metric due to its efficiency and generalization. Our extensive experiments also justify the effectiveness and robustness of this metric in evaluating the cross-channel similarity (see evidence in Appendix E) and consistently enhancing model performance.
>
> The similarity metric can be refined by incorporating domain-specific feature selection and engineering or dynamically adjusting parameters based on data distribution characteristics. These techniques are compatible and orthogonal to the proposed method, so we consider them as future work. Again, we would like to emphasize that the current metric already demonstrates the efficiency across different domains and serves as a generalizable tool to evaluate channel similarity.
>
> >How does CCM perform in real-world applications with diverse and dynamic datasets, and what adjustments are necessary to optimize its performance in such scenarios?
>
> Firstly, we clarify that **our experiments already showcase CCM’s robust performance in real-world applications with diverse datasets** (such as electricity, traffic, etc.). As mentioned in Line 246-247, CCM improves long-term forecasting performance in 90.27% of cases in MSE and 84.03% of 247 cases in MAE across 144 different experiment settings. Moreover, **CCM's versatility extends beyond scenarios where cross-channel relations remain static over time**. It supports dynamic clustering across different minibatches and time steps, which is crucial for practical applications. Due to the space limit, we discussed possible improvements such as incorporating dynamic similarity metrics or using domain-specific similarity metrics in Appendix F.
>
> >Explanations of a few cases that CCM does not outperform CI/CD strategies.
>
> As we clearly discussed in Appendix C.5, CCM method is more useful in scenarios where channel interactions are complex and significant, which is usually the case in real-world data. Therefore, in datasets where channels are nearly independent or where the inter-channel relationships are not as pronounced, the benefits of CCM's clustering mechanism may be less significant.
>
> We would like to emphasize that the average improvement rate across all cases (with different base models / datasets / forecasting lengths) is 2.443% in the long-term and 6.456% in short-term benchmarks. Specifically, CCM improves the performance in 90.27% of cases in MSE and 84.03% of cases in MAE in long-term benchmarks, given the context that the base models are already optimized for high performance. Therefore, any additional gains through CCM are noteworthy and indicative of its effectiveness in refining forecasting performance.
>
>
> We hope that the above clarification improves your confidence in our work. Let us know if you have any further questions/concerns.

---

> ### Author Response · Authors · 2024-08-12
> **We would like to hear from Reviewer yXcm**
>
> Dear Reviewer yXcm,
>
> We sincerely appreciate your comments and suggestions.  As the discussion period is close to end, we would like to follow up and ensure that our responses have adequately addressed your concerns.
>
> We would like to emphasize that **most of the issues raised were minor and had already been addressed or discussed in the original submission and appendix.** Nevertheless, we have taken this opportunity to clarify these points and further improve the revision. Specifically, we have added more discussion on the model efficiency and potentially improved similarity metric (please refer to the author rebuttal and global comments).
>
> We would be more than happy to further discuss if there are any remaining questions. Thanks again for your time and consideration.
>
> Regards, \
> Authors

---

### Official Review · Reviewer_9puR · 2024-07-12

**Soundness:** 3
**Presentation:** 3
**Contribution:** 3
**Rating:** 7
**Confidence:** 4

**Summary:**

The paper proposes a new Channel Clustering Module and a corresponding Cluster Loss to group similar channels using a cross-attention mechanism. This creates a hybrid between channel independent and channel dependent approaches. Experiments in the paper compare both with and without the proposed module on a variety of existing state of the art methods and are run on time-series datasets in different domains on both short and long-term time horizons and demonstrate that this approach is generally applicable regardless of the domain. These experiments show an improvement in performance over prior work while also enabling zero-shot forecasting. The clustering produced by this module surfaces relationships between channels which are useful for feature analysis and interpretability.

**Strengths:**

The paper is well written and easy to comprehend and has good mathematical background. The CCM and the cluster loss is well motivated and utilizes the Gumbel-softmax to represent cluster assignment in a differentiable manner. The technique is evaluated on a variety of time series datasets across diverse domains in both a short and long-term setting. This demonstrates that the method is useful across different models and generally reduces error. The computational complexity of the CCM in the inference setting is also included.

**Weaknesses:**

The paper is evaluated on a variety of real-world datasets but evaluation on some synthetic data to highlight scenarios where existing methods produce a higher error but adding the clustering module reduces the error would help solidify the claims of the paper.

**Questions:**

- In section 4.1, why shuffle channels in each batch? Channel independent implies each channel is handled individually while the motivating toy experiment here changes the channel every batch. What happens when not shuffling the channels?
- How does the prediction error change when the prediction time horizon is changed between training and inference? Is it possible to modify the time horizon after training?
- In section D.3, how do you use random and k-means for cluster assignment? Additional details about these experiments would help clarify quality improvement described in the table.
- How do you expect the performance of the CCM to change when the channel count is on the order of millions?
- What does a cluster ratio of 1.0 mean?

**Limitations:**

The authors list that the work needs to be tested on large datasets and that there is an additional performance overhead of the proposed CCM module.

---

> ### Author Rebuttal · Authors · 2024-08-07
>
> Thanks for your valuable feedback and positive comments.  We address the potential concerns as follows.
>
> >Adding evaluation on synthetic dataset will help solidify the claims.
>
> While synthetic datasets serve their purpose in controlled experiments, our study's emphasis on real-world datasets is crucial and more reasonable for validating the practical applicability and robustness of the proposed CCM. Unlike synthetic datasets generated in previous studies, which may not encompass the full spectrum of real-world complexities, we leverage diverse real-world benchmarks. These datasets inherently capture intricacies such as varied data distributions, noise levels, and nuanced interdependencies between channels in real-time forecasting scenarios. This focus allows us to demonstrate CCM's effectiveness in addressing practical challenges across different domains, highlighting its superiority over synthetic benchmarks that often fail to replicate such complex interactions.
>
> >Why shuffle channels in each batch? What happens when not shuffling the channels?
>
> Shuffling channels aim to remove channel identity information. In the toy experiment in Sec 4.1, we train the model in two patterns: A) the model is trained on the original dataset and B) The model is trained on the randomly shuffled dataset.  Let $W_i$ represent the weights for the $i$-th channel. In Pattern A, $W_i$ is optimized upon the $i$-th channel only, while in Pattern B, the optimization of $W_i$ will lose channel identity information, which causes performance degradation on both CI and CD models (from Table 1). Therefore, this toy experiment justifies that channel identity information benefits model performance in general.
>
> >How does the prediction error change when the prediction time horizon is changed between training and inference? Is it possible to modify the time horizon after training?
>
> Yes, it is possible to adjust the forecasting horizon after training, although it's not typically done in practice. When shortening the forecasting horizon post-training, the prediction error remains relatively stable because the training loss averages errors over each future step. However, if extending the horizon (e.g., from H1 to H2 where H2 > H1), the conventional approach involves forecasting H1 steps first, then using that prediction as input to forecast the subsequent H2 - H1 steps. This sequential process tends to increase prediction error due to accumulated inaccuracies over the extended horizon.
>
> >How do you use random and k-means for cluster assignment?
>
> In the Random method, each channel is assigned to clusters with uniform probability. We use *sklearn.cluster.KMeans* to implement the k-means algorithm with a default maximum number of iterations (*max_iter*=300). The input channel embeddings remain the same as that of our proposed Cluster Assigner. We keep the number of clusters consistent for three clustering methods in the ablation study. We will add details in our revised version.
>
> >How do you expect the performance of the CCM to change when the channel count is on the order of millions?
>
> Our clustering technique, which scales linearly with respect to channel counts, has been tested to efficiently handle large datasets. Numerous experiments across diverse datasets, including large-scale datasets such as traffic (15,122,928 observations) and stock (13,900,000 observations), provide strong evidence for CCM's promising scalability on large-scale datasets: 1) CCM consistently reduces model complexity for CI models (e.g. DLinear and PatchTST), regardless of dataset scale or size (Fig. 5); 2) The linear scaling w.r.t channel count enables efficient handling of high-dimensional data, crucial for many practical applications; and 3) our experiments reveal that the performance gains achieved by CCM are maintained across different dataset sizes, suggesting that the method's benefits are robust and not limited to specific data scales. Therefore, CCM is supposed to effectively group channels based on similarities while maintaining optimal computational efficiency.
>
> >What does a cluster ratio of 1.0 mean?
>
> A cluster ratio of 1.0 indicates that the number of clusters equals the number of channels, but it does not imply channel independence. In our experiments, even with a cluster ratio of 1.0, we observed clustering phenomena in channel embeddings, albeit with some empty clusters. This ratio is maintained for experimental rigor and to validate the robustness of the model.

---

> > ### Comment · Reviewer_9puR · 2024-08-13
> >
> > Thank you for the rebuttal. These have addressed most of my concerns.

---

> ### Author Response · Authors · 2024-08-12
> **We would like to hear from Reviewer 9puR**
>
> Dear Reviewer 9puR,
>
> As the discussion period is close to end, we would like to follow up and ensure that our responses have adequately addressed your concerns. We sincerely appreciate your comments and suggestions, which have significantly contributed to improving our paper. We would be more than happy to further discuss if there are any remaining questions. Thanks again for your time and consideration.
>
> Regards,
> Authors

---

### Author Response · Authors · 2024-08-08
**Rebuttal Summary by Authors**

Dear Reviewers,

We sincerely appreciate your thorough and valuable feedback on our manuscript. We have provided detailed clarifications and additional experimental results to address all concerns raised.

## I. Contributions
We are pleased that many reviewers have offered positive comments, and we would like to highlight our key contributions:

- **Generalization and Comprehensive Evaluation:** CCM is a model-agnostic channel strategy that is adaptable to most mainstream time series models. Experiments are thoroughly conducted across multiple different models, datasets, forecasting scenarios. [Reviewers 9puR, rxdY, SRyH]
- **Superior Performance Improvement:** Our extensive experiments demonstrate CCM's effectiveness in significantly enhancing base model performance for time series forecasting across multiple benchmarks. [Reviewers 9puR,  rxdY]
- **Zero-shot Capability:**  CCM supports zero-shot forecasting by leveraging learned prototypes, which is crucial for handling challenging scenarios with scarce or limited training data. [Reviewers yXcm, rxdY]
- **Clear Motivation and Novelty:** CCM addresses limitations of existing strategies (CD and CI) on suboptimal performance or insufficient channel interactions, through a novel approach to channel clustering for time series forecasting. We appreciate the reviewers' acknowledgment of our manuscript's clarity and readability. [Reviewers 9puR, SRyH]

## II. Common Question
We also recognize that there are some common questions from reviewers. We conclude the important clarifications as below.

> “CCM’s scalability to extremely large datasets remains to be tested” as mentioned in the Limitation section.

- We included **theoretical analysis (Sec 4.3)** and **empirical test (Appendix C.6)** for CCM’s complexity and efficiency. The complexity is linear w.r.t. number of channels. The modest computational overhead on CD models is reasonable and acceptable, as justified by the significant performance improvements achieved. It is worth noting that CCM consistently reduces the model complexity of CI models.
- Our current research prioritizes time series benchmarks used in prior works [1,2,3,4]. Our experiments include a traffic dataset comprising 15,122,928 observations and a stock dataset comprising 13,900,000 observations. These datasets represent large-scale real-world datasets widely used in previous research within this domain.
- Scalability should not be an issue of CCM in real-world deployment due to several empirical evidence:
    * Consistent reduction in model complexity for CI models
    * Linear scaling with respect to the number of channels, enabling efficient handling of high-dimensional data
    * Maintained performance gains across different dataset sizes


## III. Main modifications in our revised version
According to reviewers’ suggestions, we have made the following improvements in our revised version:
- We added implementation details of the k-means algorithm in ablation studies in Appendix D.3.
- We provided significance tests with p-value for cases where values are close before/after applying CCM.
- We conducted ablation studies on the parameter $\beta$ in the loss function to demonstrate CCM’s robustness.

We hope these modifications and clarifications address your concerns while highlighting the contributions and significance of our work. We are confident that CCM represents a valuable advancement in time series forecasting and look forward to further constructive discussions. Thank you for your time and consideration.


Sincerely,\
Authors



[1] TSMixer: An All-MLP Architecture for Time Series Forecasting\
[2] Are Transformers Effective for Time Series Forecasting?\
[3] A Time Series is Worth 64 Words: Long-term Forecasting with Transformers\
[4] TimesNet: Temporal 2D-Variation Modeling for General Time Series Analysis

---

### Decision · Program_Chairs · 2024-09-25

**Decision:**

Accept (poster)

**Comment:**

This paper proposed to model the correlation between channels to improve  time series forecasting via an attention-based channel clustering method. Experimental results have demonstrated the effectiveness of the proposed channel clustering approach. Although the channel clustering approach is new, the channel dependency modeling  is not the first time explored in literature, e.g., [1-2]. A detailed discussion is necessary.

[1] Rethinking Channel Dependence for Multivariate Time Series Forecasting: Learning from Leading Indicators

[2] Enhancing Multivariate Time Series Forecasting with Mutual Information-driven Cross-Variable and Temporal Modeling

Some reviewers have concerns on the computation complexity and scaling ability. Authors have provided responses to these questions.